# Consummating ion desolvation in hard carbon anodes for reversible sodium storage

Ziyang Lu[1,6], Huijun Yang [1,6], Yong Guo[2], Hongxin Lin[3], Peizhao Shan[3], Shichao Wu [2], Ping He [4], Yong Yang [3], Quan-Hong Yang [2] ✉ & Haoshen Zhou [1,5] ✉

Hard carbons are emerging as the most viable anodes to support the commercialization of sodium-ion (Na-ion) batteries due to their competitive performance. However, the hard carbon anode suffers from low initial Coulombic efficiency (ICE), and the ambiguous Na-ion (Na$^+$) storage mechanism and interfacial chemistry fail to give a reasonable interpretation. Here, we have identified the time-dependent ion pre-desolvation on the nanopore of hard carbons, which significantly affects the Na$^+$ storage efficiency by altering the solvation structure of electrolytes. Consummating the pre-desolvation by extending the aging time, generates a highly aggregated electrolyte configuration inside the nanopore, resulting in negligible reductive decomposition of electrolytes. When applying the above insights, the hard carbon anodes achieve a high average ICE of 98.21% in the absence of any Na supplementation techniques. Therefore, the negative-to-positive capacity ratio can be reduced to 1.02 for full cells, which enables an improved energy density. The insight into hard carbons and related interphases may be extended to other battery systems and support the continued development of battery technology.

Compared to lithium (Li), sodium (Na) reserves in the earth's crust are dozens of times higher, making sodium-ion (Na-ion) batteries a more cost-effective and sustainable alternative to lithium-ion (Li-ion) batteries[1,2]. Among the potential anode materials, hard carbon has garnered considerable attention owing to its competitive advantages in abundance, capacity, and operating potential[3]. However, a major hurdle to the commercialization of hard carbon anodes is the low initial Coulombic efficiency (ICE), which is much lower than the case of graphite anodes in Li-ion batteries[4,5].

The low ICE of the hard carbon anode is generally attributed to irreversible Na loss caused by electrolyte decomposition during the solid electrolyte interphase (SEI) formation[6,7], which is always relevant to defect concentration and specific surface area as it largely affects the decomposition of electrolytes[8–10]. However, the theory of SEI formation cannot fundamentally explain some experimental aspects. For instance, using hard carbon and graphite with the same specific surface area, the ICE of hard carbons in Na-ion batteries is significantly inferior to graphite anodes in Li-ion batteries[11,12]. In contrast, hard

[1]Graduate School of System and Information Engineering, University of Tsukuba, Tsukuba, Japan. [2]Nanoyang Group, Tianjin Key Laboratory of Advanced Carbon and Electrochemical Energy Storage, School of Chemical Engineering and Technology, Collaborative Innovation Center of Chemical Science and Engineering (Tianjin), Tianjin University, Tianjin, P. R. China. [3]State Key Laboratory for Physical Chemistry of Solid Surfaces, Collaborative Innovation Center of Chemistry for Energy Materials, and Department of Chemistry, College of Chemistry and Chemical Engineering, Xiamen University, Xiamen, P. R. China. [4]Center of Energy Storage Materials & Technology, College of Engineering and Applied Sciences, Jiangsu Key Laboratory of Artificial Functional Materials, National Laboratory of Solid State Micro-structures, and Collaborative Innovation Center of Advanced Micro-structures, Nanjing University, Nanjing, P. R. China. [5]Present address: Center of Energy Storage Materials & Technology, College of Engineering and Applied Sciences, Jiangsu Key Laboratory of Artificial Functional Materials, National Laboratory of Solid State Micro-structures, and Collaborative Innovation Center of Advanced Micro-structures, Nanjing University, Nanjing, P. R. China. [6]These authors contributed equally: Ziyang Lu, Huijun Yang. ✉e-mail: qhyangcn@tju.edu.cn; hszhou@nju.edu.cn

carbons with high specific surface area (over 600 m² g⁻¹ by MgO-template techniques) have been found to exhibit well-defined pore-filling plateaus and high ICE values (88%) that are not predicted by current theories[13]. Furthermore, the dominant mechanistic model of Na-ion (Na⁺) storage does not stand up to scrutiny when applying the current SEI-related theories. In addition to the ICE issues, there are also many conflicts between existing Na⁺ storage models and experimental results in hard carbon anodes.

The dominant Na⁺ storage model divides the discharge curve into two regions: the slope and plateau region, which are believed to correspond to two Na⁺ storage modes[14]. The former is the graphitic domain, which is formed by the random stacking of few-layer graphenes and is generally attributed to the insertion of Na⁺ in the stacked graphene or adsorption on the defective surface, contributing to the slope capacity[15–18]. The latter is the so-called "ultra-micropores" or nanopores, which can be seen as voids among the randomly stacked graphitic domains. In this region, pore filling occurs due to its unique ultra-microporous structure to accommodate Na clustering, which corresponds to the plateau capacity and governs the charge-discharge process[19,20]. This prevailing theory believes that solvated Na⁺ cannot enter the nanopore due to the narrow pore entrance and the electrolyte merely decomposes on the surface of hard carbons to form SEIs[21]. However, this theoretical premise fails to provide reasonable explanations for many experimental phenomena. For instance, both the interfacial resistance across the SEI and desolvation resistance for the pore filling are significantly smaller than the Na⁺ insertion/adsorption process, which is completely different from the Li⁺ storage in the graphite[22,23]. Furthermore, the above model fails to reasonably explain the Na⁺ transport behavior of the pore-filling process and the shrink of the nanopore with cycling[19,20]. Addressing these issues requires a new theoretical framework that can account for the unique properties of hard carbons and provide a more comprehensive understanding of the fundamental processes involved in Na⁺ storage.

Through experimental characterization combined with modeling, we have identified an SEI-independent pre-desolvation process occurring on the nanopores of hard carbon anodes, driven by capillary effect and osmotic pressure. The solvation structure evolution during the pre-desolvation process has a significant impact on the reversibility of Na⁺ storage, which is distinguished from the desolvation effect on

microporous carbon in supercapacitors[24,25]. Our model shows that hard carbon anodes intrinsically induce a time-dependent desolvation process and enable a dual-SEI structure both on the surface and inside of nanopores. Our demonstrations correlate between the SEI-independent desolvation and the Na storage efficiency, providing a more comprehensive explanation for the above-mentioned phenomena. A high average ICE of 98.21% (standard deviation: 1.55%) can be reached by simply prolonging the aging time to achieve sufficient desolvation, which is significantly higher than the recently reported ICE data (generally below 92%) using advanced hard carbons and optimized electrolytes[26–34]. To compensate for the low ICE, the negative/positive (N/P) capacity ratio is generally over 1.2 for previously reported full cells. Benefiting from the high ICE obtained in this work, the N/P ratio can be reduced to 1.02, which enables a high energy density of 282 W h kg⁻¹ for the full cell. Using the dual-SEI model and element valence deep profile analysis, we have determined that irreversible Na losses are caused by Na⁺ depletion in dual-SEI formation and Na⁺ being trapped in graphene sheets. The proposed model will advance our understanding of the Na⁺ storage mechanism in hard carbon anodes and facilitate its practical application.

## Results

The typical hard carbon sphere (Supplementary Figs. 1 and 2) was used as a model material to study the correlation between ICE and aging time. As shown in Fig. 1a, by simply increasing the aging time, the ICE of the Na||hard carbon half-cell using an ether electrolyte (1 M NaPF₆-diglyme, G2) gradually increases. Note that, the aging process is carried out after the cell is assembled, and the cell test follows the aging. Upon resting for 12 h, the ICE is 90.14%. As the aging time increases to 10 days, the ICE significantly improves and shows the highest value of 99.09% (Fig. 1b), and the average ICE reaches 98.21% (standard deviation: 1.55%) at a specific current of 20 mA g⁻¹. Since ICE is affected by the test conditions and electrode preparation process, the related detailed experimental process is presented in the experimental section. Supplementary Table S1 compares the recently reported ICE data of hard carbon under corresponding test conditions (including electrolytes, binders, separators, and specific currents). Notably, the ICE achieved in this work is the highest value reported to date in the absence of any Na supplementation techniques. For thick electrodes

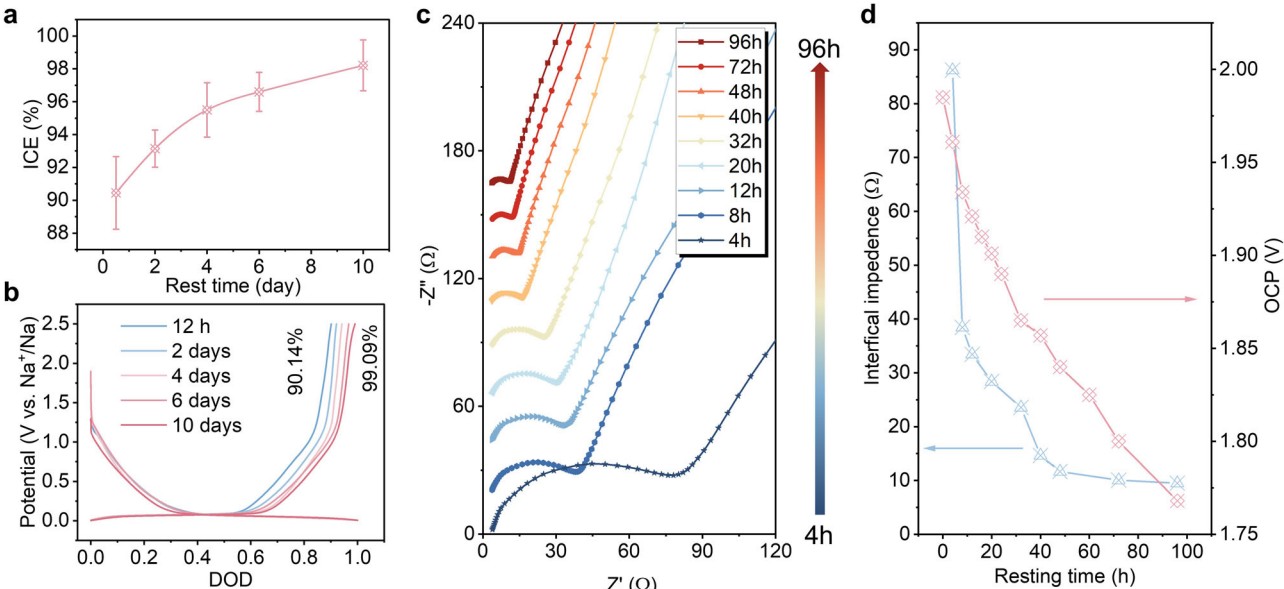

**Fig. 1 | Correlation between ICE of Na⁺ storage in hard carbon anodes and aging time. a** The curve of ICE vs resting time. **b** The initial charge−discharge curve at different resting times. **c** Nyquist plots of Na||hard carbon half cells recorded at different aging times. **d** Summary of interface impedance and OCP changes at different aging times.

with high mass loading (13.7 mg cm$^{-2}$), the ICE can also see an increase with aging time (Supplementary Fig. 3). After aging for 10 days, an average ICE of 97.17% can be reached. We further investigated the case of commercial hard carbons. As shown in Supplementary Fig. 4, the ICE also gradually increases with aging time. An average ICE of 97.67% can be obtained after aging for 10 days, which further verifies the impact of aging time on ICE. To untangle the underlying causes of the high ICE behind aging, the interface resistances under different aging times were systematically examined. As shown in Fig. 1c, d, the impedance value is 86.2 Ω after resting for 4 h, and we can see a sharp reduction after aging for 40 h (14.6 Ω). The fitting results are shown in Supplementary Table S2. Subsequently, the impedance value decreases steadily with the aging time. Correspondingly, the open circuit potential (OCP) gradually decreases with aging time (Fig. 1d), which is a typical characteristic of ICE improvement[35]. In general, the ICE and interface impedance would not be significantly influenced by aging time for plate electrodes or plate-like electrodes. However, the electrochemical behavior of porous electrodes is always related to the standing time, which needs to be completely wetted by sufficient aging.

## The chemical origin of high ICE with aging

Subsequently, the porous feature of hard carbon was systematically analyzed. To accurately confirm the pore size of hard carbons, 3 A and 4 A zeolite molecular sieve and zeolitic imidazolate framework-7 (ZIF-7) with well-defined pore size were also tested by $CO_2$ adsorption measurements (Fig. 2c). All four materials show type I adsorption isotherms, demonstrating a typical micropore feature[35]. The ZIF-7 and 3 A zeolite molecular sieves have effective pore sizes of 2.9 Å and 3.2 Å,

respectively, which are much smaller than 4 A zeolite molecular sieves. The ZIF-7 and 3 A and 4 A zeolite molecular sieves show almost coincided adsorption and desorption isotherms, demonstrating that there is no capillary condensation phenomenon. But for the hard carbon, it shows an H2-type hysteresis loop, which is a typical feature of ink-bottle-like pores. This kind of pore has greater filling resistance than a straight pore, taking a longer time to wet and fill the nanopore. Therefore, it has a pore size of ~4 Å, but its $CO_2$ adsorption capacity is lower than 4 A zeolite molecular sieve, and the 4 Å pore was used to simulate the desolvation process. As shown in Supplementary Fig. 5, two regions were selected to calculate the interlayer spacing, and the average values are 3.94 Å and 4.05 Å, respectively. It is basically consistent with the interlayer spacing calculated from the Bragg equation (0.397 nm). Note that, some regions of stacked graphene layers can also be regarded as orifices of nanopores due to the short-range ordered structural features of hard carbons, which might also be detected by $CO_2$ adsorption measurements. The hard carbon exhibits a high skeletal density of 2.183 g cm$^{-3}$ probed by He pycnometry experiments, which is slightly lower than the natural graphite (2.26 g cm$^{-3}$)[36], indicating that only a small amount of internal, inaccessible porosity exists. Recent studies have shown that these inaccessible closed ultra-micropores can also effectively contribute to reversible capacity, especially the plateau capacity[37,38], and there are differences in the storage efficiency of Na$^+$ for different types of closed ultra-micropores[39,40]. Therefore, those ultra-micropores that are inaccessible for $N_2$ and $CO_2$ probe molecules can be detected by He molecules with smaller sizes. It is directly related to subsequent ion desolvation. The filling dynamics are well elucidated by Olivier Vincent et al.[41,42]. The driving force for filling comes from two aspects,

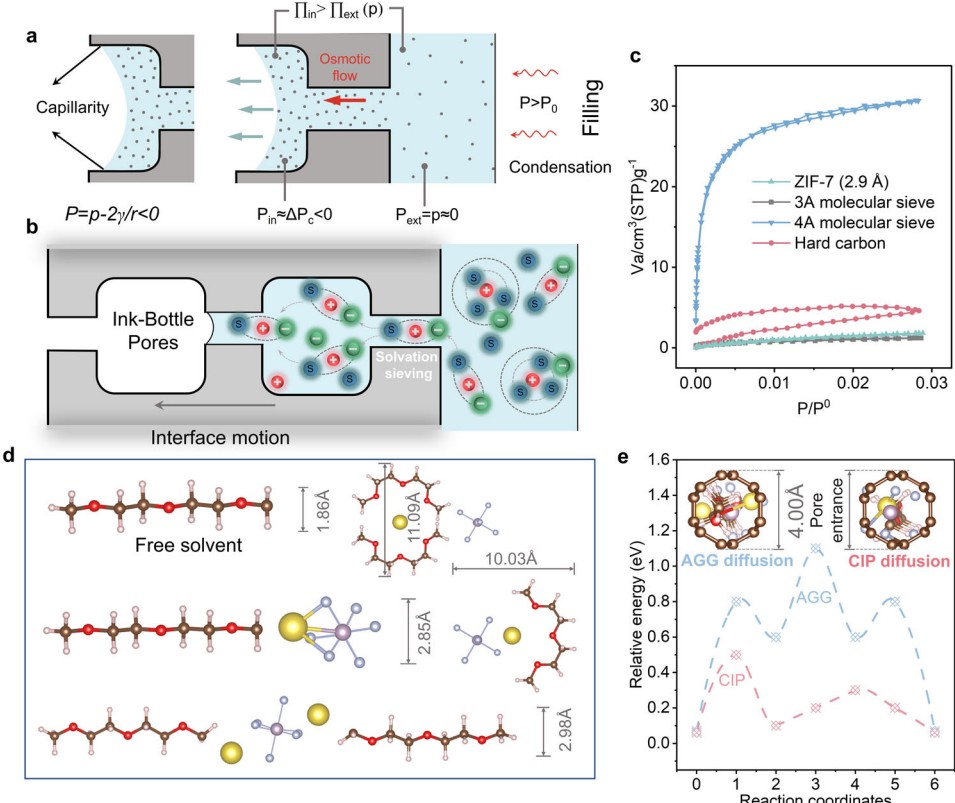

**Fig. 2 | SEI-independent pre-desolvation on the nanopore of hard carbons.**
**a** Electrolyte filling process in hard carbon nanopores. **b** Phase equilibrium when considering the combined impacts of capillary stresses (Kelvin−Laplace) and solution (osmotic equilibrium) by adapting classical thermodynamical treatments. **a**, **b** are adapted with permission from ref. 41. Copyright 2019, American Chemical Society. **c** $CO_2$ adsorption/desorption isotherms of ZIF-7, 3 A zeolite molecular sieve, 4 A zeolite molecular sieve, and hard carbons. **d** The sizes of various solvation structures are calculated by DFT calculation. **e** The energy barrier of the solvation structure of CIP and AGG crosses the hard carbon nanopores with a pore size of 4 Å. Insets are sectional views of CIP and AGG crossing the bottleneck of hard carbons.

namely osmotic pressure (Eq. 1) and hydrostatic force (Eq. 2). $p$ and $p_s$ present the equilibrium vapor pressure of pure solvents and solutions, respectively. $\nu_w^{liq}$ is the molar volume fraction of pure solvent. $\prod_{ext}$ and $\prod_{in}$ are the osmotic pressures outside and inside the pores. According to Eq. 1, $p > p_s$ corresponds to the local equilibrium of the steam, the external solution is diluted by the condensation of the steam. This osmotic pressure causes the solvent to permeate from the outside of the pore to the inside, and this osmotic flux persists even if the nanopore is not fully permeable to the solute. Hydrostatic force is actually a capillary force, which always favors the filling of the pores and leads to increased concentration in the pores (Fig. 2a, b). Based on these two driving forces, the liquid flow rate in the nanopore can be expressed as Eq. 3. Sigma ($\sigma$) is a dimensionless reflection coefficient related to the solute radius ($r_s$) and pore radius ($r_p$). Combined with Eqs. 3 and 4, it can be seen that the smaller the radius of the hole and the larger the solute radius, the smaller the filling flow rate. Therefore, it takes a long time for the electrolyte to be fully filled in nanopores, which significantly affects the interfacial impedance of hard carbon anodes.

$$\Delta \prod = \prod_{ext} - \prod_{in} = p - p_s - \frac{RT}{\nu_w^{liq}} \ln\left(\frac{p}{p_s}\right) \approx -\frac{RT}{\nu_w^{liq}} \ln\left(\frac{p}{p_s}\right) \quad (1)$$

$$\Delta P = P_{ext} - P_{in} = p - p_s - \Delta P_c \approx -\Delta P_c > 0 \quad (2)$$

$$Q = \lambda \frac{k}{\eta} \left(\Delta P - \sigma \Delta \prod\right) \quad (3)$$

$$\sigma = \left[1 - \left(1 - \frac{r_s}{r_p}\right)^2\right]^2 \quad (4)$$

These issues are easier to understand from the perspective of computational chemistry and electrolyte sieving chemistry. Since the size of the solute would significantly affect the flow rate (Eqs. 3 and 4), and the solute often exists in the solvated form in the actual electrolyte system, we calculated the size of several common solvation structures instead of pure solutes. To accurately reflect the structure of hard carbon, it is often necessary to establish a very fine and large-scale model[43,44]. However, it is usually time-consuming and computationally intensive. Since the pore size of hard carbon is the key factor affecting the desolvation process[45,46], a graphene ring with a diameter of 4 Å was applied for simplifying simulations and calculations (Fig. 2e). The method of constructing models by grasping the key influencing factors to ensure its rationality while mitigating computational effort is widely used[47–49]. For the solvent-separated ion pair (SSIP) structure that dominates dilute solutions with size greater than 10 Å (Fig. 2d), it is blocked by the nanopore (Fig. 2b). However, for the contact ion pair (CIP) and aggregate (AGG) with suitable orientation, it is possible to pass through the nanopore, although there are still maximum barriers of 0.5 eV and 1.1 eV, respectively (Fig. 2e, d and Supplementary Fig. 6). The diffusion barrier of CIPs and AGGs is counteracted by capillary force and osmotic pressure. In fact, pore filling and pre-desolvation occur simultaneously and tend to improve over time. This size-based pre-desolvation process behaves similarly to what has been observed in metal-organic frameworks (MOFs) and zeolite molecular sieves[46,50]. However, the difference is that the hard carbon with intrinsic nanopores itself acts as both the host for pre-desolvation and the carrier for the charge storage, which may be an important factor in explaining why only hard carbon can achieve efficient Na$^+$ storage among numerous carbon materials.

Note that, conventional ion desolvation converts solvated cations into bare cations via electronically insulating but ionically conductive SEIs. However, the pre-desolvation on the nanopore of hard carbons only partially removes the solvated shell rather than transforming it into bare Na$^+$. The unique pre-desolvation based on the nanopore was identified by Raman, Fourier-transform infrared spectroscopy (FT-IR), and nuclear magnetic resonance (NMR) spectra. As shown in Fig. 3a, the peak located at about 740 cm$^{-1}$ can be attributed to the symmetric P–F stretching of PF$_6^-$ (A1g)[51,52], which can be observed for all electrolytes apart from the pure solvent. As the concentration increases, the solvation structure gradually evolves from SSIPs or CIPs to AGGs with higher aggregation, which is a typical feature of improved interaction between the PF$_6^-$ and Na$^+$[53]. Interestingly, the Raman spectra of hard carbons fully aged in the 1 M electrolyte show a larger blue shift and higher peak intensity than the 2 M electrolyte, which is close to the saturated state. In this case, AGGs dominate the solvation configuration. The evolution of the solvation configuration can also be identified from the CH$_2$ wagging mode of solvents[54]. For G2 solvents (Fig. 3b), the CH$_2$ wagging mode of the free solvent peak can be observed at 844.8 cm$^{-1}$. However, as the concentration increases, the free solvent peak gradually diminishes while the coordinated solvent peak becomes more prominent. For the electrolyte in the nanopore of hard carbons, the signal is entirely taken up by the coordinated solvent, leaving no room for free solvents. The function of hard carbon in converting solvation structures can also be observed from CH$_2$ rocking and C–O–C stretching of G2 solvents[53,55] as shown in Supplementary Fig. 7. The solvation structure in the nanopore is further verified by $^{23}$Na NMR spectra (Fig. 3c). The 1 M liquid electrolyte exhibits a sharp peak at −2.06 ppm with a narrow peak width. Nonetheless, the electrolyte in hard carbons, the signal shows a notable increase in peak width. This can be attributed to the electrolyte trapped in the nanopore, which hinders the diffusion of Na$^+$[56]. When a spinning speed of 25 kHz is applied, the peak shifts to the upfield, indicating an increased electron density of solvated Na$^+$. This can be attributed to the rotation that accelerates the pre-desolvation process. Moreover, hard carbon possesses some degree of conductivity and generates eddy currents, which may result in the peak shift towards the upfield. Supplementary Fig. 8 clearly illustrates the pre-desolvation achieved by the nanopore of hard carbons. Then, the solvation structure in the nanopore was studied through molecular dynamic (MD) simulations. For the dilute electrolyte, salt is completely dissociated and Na$^+$ is closely coordinated with G2 solvents surrounded by PF$_6^-$ to form SSIP in most cases (Fig. 3d). However, for electrolytes in hard carbon nanopores, the snapshot at an equilibrium trajectory presents a polymerization network connected by PF$_6^-$ and Na$^+$ (Fig. 3e). Each PF$_6^-$ anion coordinates with multiple Na$^+$, forming AGGs and partially CIPs, and resulting in an increase in Na$^+$-P radial distribution function (Supplementary Fig. 9). These results are consistent with the Raman and FT-IR analyses.

Furthermore, we analyzed the time-dependent solvation structure evolution in hard carbons by in situ Raman spectroscopy (Fig. 3f). The changes of solvation structures can be observed from the increasing intensity of A1g of PF$_6^-$, C–O–C stretching and CH$_2$ rocking of solvents. In addition, the A1g peak of PF$_6^-$ shifts towards high wavenumbers because of the increased interaction between PF$_6^-$ and Na$^+$. The C–O–C stretching and CH$_2$ rocking-related peaks are blue-shifted due to the ion-dipole attraction between dissociated Na$^+$ and solvents. Furthermore, the aging duration would result in varying depths of electrolyte diffusion in hard carbons, which can be detected via X-ray photoelectron spectroscopy (XPS) with depth etching function. For the hard carbon immersed in typical 1 M NaPF$_6$-G2 electrolytes (without Na metal) for 10 days, the Na 1$s$ peak at 1072.4 eV corresponding to ionic Na can always be detected and the peak intensity remained unchanged during the 21-min etching (Fig. 3g). The solvation structure related signals can also be observed from F 1$s$ (Supplementary Fig. 10) and time-of-flight secondary ion mass spectrometry (ToF-SIMS) mapping of CsNa$^+$ and Cs$_2$F$^+$ (Fig. 3i, j and Supplementary Fig. 11). However, for the case that hard carbon immersed

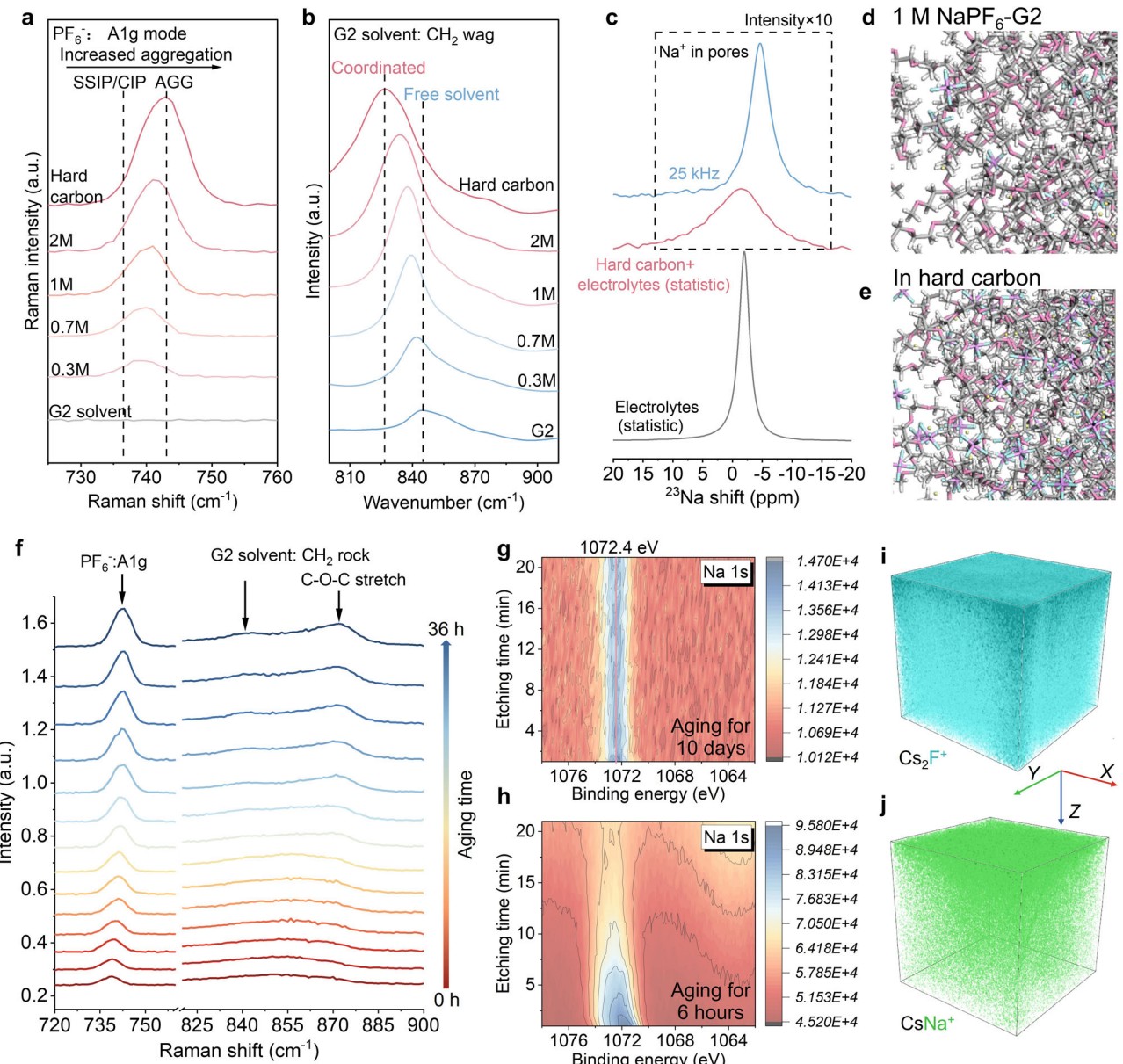

**Fig. 3 | Analysis of solvation structure evolution in hard carbons. a, b** Raman (**a**) and FT-IR (**b**) spectra of electrolytes with different concentrations and the electrolyte configuration in the nanopore of hard carbons. **c** $^{23}$Na NMR spectra of pristine liquid electrolytes and hard carbon modified electrolytes. **d, e** Molecular dynamics simulation of the 1 M NaPF$_6$-G2 electrolyte (**d**) and the electrolyte inside hard carbons (**e**). **f** Raman spectra of electrolytes in hard carbons with different aging times. **g, h** XPS spectra of Na 1$s$ with different etching times (0–25 min) collected from the hard carbon immersed in 1 M NaPF$_6$-G2 electrolytes (without Na metal) for 7 days (**g**) and 12 h (**h**). **i, j** ToF-SIMS mapping of CsNa$^+$ (**i**) and Cs$_2$F$^+$ (**j**) for hard carbon immersed in ether electrolytes (without Na metal).

in electrolytes for 6 h, the solvation structure diffuses only within the shallow layer of hard carbons. Therefore, the intensity of the Na 1$s$ signal gradually weakens with etching (Fig. 3h). This significant difference caused by aging time obviously cannot be explained by previous theories and models. However, the aforementioned phenomena can be well interpreted when the porosity of hard carbons and the associated pre-desolvation process are taken into account. The aggregation of the electrolyte in the nanopore is greatly enhanced following pre-desolvation through the nanopores of hard carbons. The increased aggregation favors the participation of PF$_6^-$ in the solvation sheath and reduces the Na$^+$-driven electron-withdrawing power. Therefore, the LUMO (the lowest unoccupied molecular orbital) is significantly reduced, which effectively curbs the electrolyte decomposition and ultimately leads to improved ICE[57]. Any process relating to electrochemical reactions or charge transfers at the electrode

surface involves ion desolvation[58,59], which is also discussed for microporous carbons in supercapacitors[25]. Although it was established that pore size can affect the charge storage behavior of supercapacitors, the associated solvation evolution was not revealed[24,25,60]. In addition, the unique ultra-microporous structure of hard carbons makes its desolvation process and effects such as the impact on ICE and SEI formation completely different from those of microporous carbon in supercapacitors.

## Dual-SEI model and the analysis of irreversible Na loss

The SEI-independent pre-desolvation on the hard carbon will inevitably generate a new SEI inside the nanopore (I-SEI). It is completely different from the SEI formed on the surface (S-SEI). Therefore, the dual-SEI model is naturally proposed (Fig. 4a). In order to better identify these two kinds of SEIs, we compared the SEIs formed in both

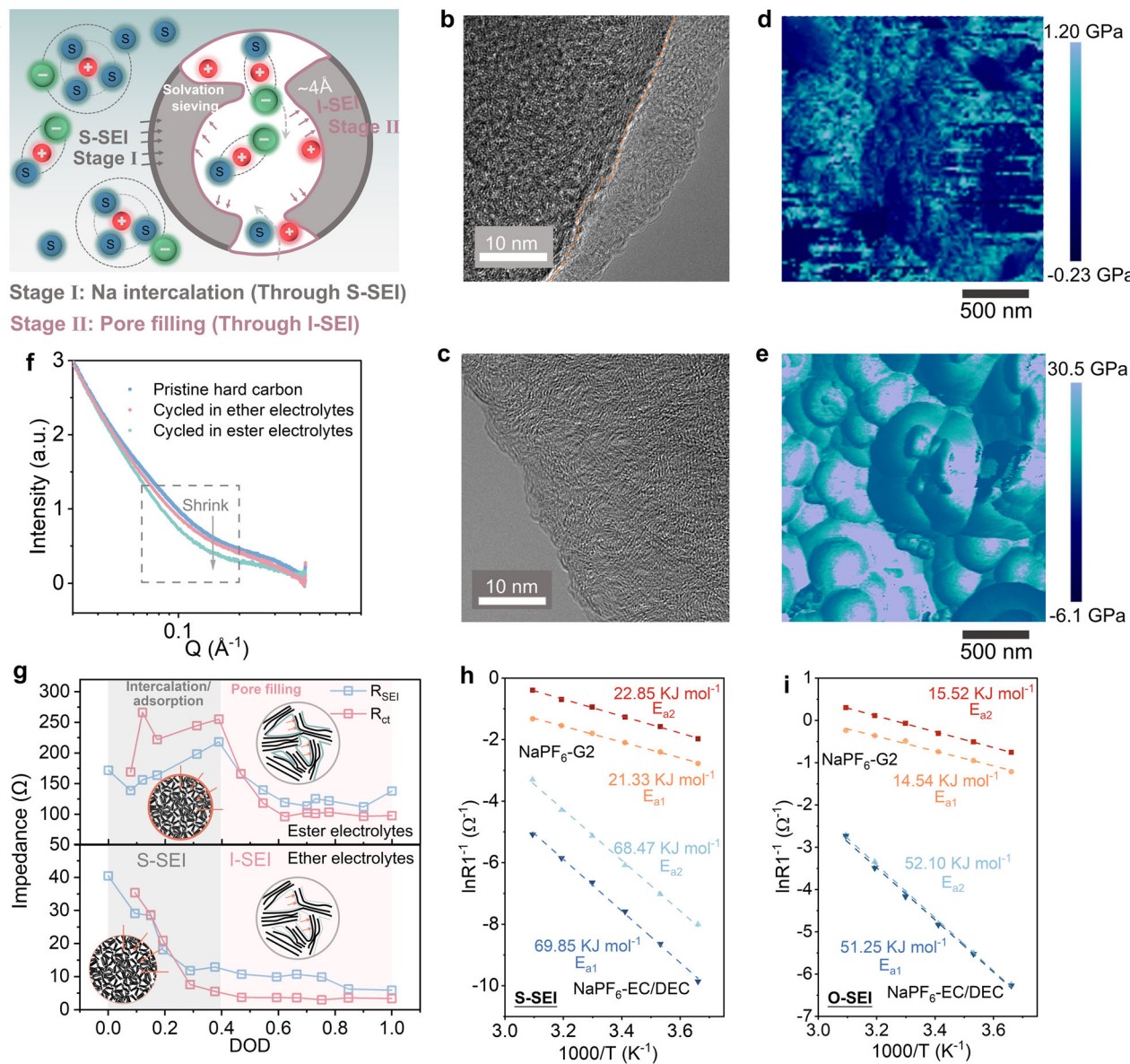

**Fig. 4 | Validation of the dual-SEI model. a** Schematic of pre-desolation and dual-SEI formation process on hard carbons. **b**, **c** TEM images of hard carbon after 10 cycles in 1 M NaPF$_6$-EC/DEC electrolytes (**b**) and 1 M NaPF$_6$-G2 electrolytes (**c**). **d**, **e** Two-dimensional AFM maps of elastic modulus AFM analysis of cycled hard carbons in 1 M NaPF$_6$-EC/DEC electrolytes (**d**) and 1 M NaPF$_6$-G2 electrolytes (**e**). **f** SAXS patterns of pristine hard carbons and cycled hard carbons in ester and ether electrolytes. **g** The $R_{SEI}$ and $R_{ct}$ at different discharge states in ester electrolytes and ether electrolytes. **h** Arrhenius plots were obtained at a fully charged state. **i** Arrhenius plots obtained at fully discharged state.

ester (1 M NaPF$_6$-EC/DEC) and ether electrolytes (1 M NaPF$_6$-G2). For the S-SEI formed in ester electrolytes, the thickness is about 10 nm after 10 cycles (Fig. 4b), which can be identified from transmission electron microscopy (TEM) images. However, the S-SEI formed in ether electrolytes is very thin and even negligible due to the excellent reductive stability (Fig. 4c). In addition to the difference in SEIs, significant differences can also be seen for the structure of the hard carbon harvested from different electrolytes. As shown in Supplementary Fig. 12, the hard carbon cycled in ester electrolytes shows much larger interlayer spacing and shorter fringe length compared to the case using ether electrolytes, which originates from the irreversible intercalation of Na$^+$ expanding the graphene sheets and interrupting the original continuous layered structure of hard carbons. The irreversible Na$^+$ intercalation and SEI formation are responsible for the low ICE in ester electrolytes, which was confirmed in the subsequent XPS analysis. Then, the mechanical property of S-SEIs was investigated by

atomic force microscopy (AFM). As shown in Fig. 4d, e, the elastic modulus of the S-SEI generated in 1 M NaPF$_6$-G2 is much higher than that formed in ester electrolytes as demonstrated by the two-dimensional AFM mapping. Then, eight points were selected on each mapping to quantify its modulus and mechanical characteristics (Supplementary Fig. 13). The average elastic modulus of the eight sites for the S-SEI generated in ether electrolytes is 13.12 GPa, which is significantly higher than that formed in ester electrolytes (0.48 GPa). The difference in mechanical properties can be attributed to the difference in SEI compositions. Later, we will discuss this in detail.

For the I-SEI, it can be detected by small-angle X-ray scattering (SAXS). After 10 cycles, the pore volume of hard carbon shrinks in both electrolytes (Fig. 4f). However, the shrinking of the pore volume is more severe in ester electrolytes, indicating that a thicker SEI is formed in the nanopores. Therefore, F and Na, the main constituent elements of the SEI, are not only detected in the surface layer but also in the bulk

(Supplementary Fig. 14). The difference in compositions and properties of these two SEIs would exhibit completely different interfacial transport behaviors, which is verified by in situ electrochemical impedance spectroscopy (EIS). The ICE reaches 96.4% for ether electrolytes during in situ EIS tests, which is much higher than that in ester electrolytes (82.8%) (Supplementary Fig. 15). Moreover, the significant ICE difference in ester and ether electrolytes allows us to deduce the irreversible Na loss from the electrochemical behavior of hard carbons. Both the $Na^+$ transport resistance through the SEI ($R_{SEI}$) and the charge transfer resistance ($R_{ct}$) in ether electrolytes are significantly smaller than those in ester electrolytes during the discharging process (Fig. 4g). The fitting results are shown in Supplementary Tables S3 and S4. The $R_{ct}$ and $R_{SEI}$ for the slope region are significantly lower than the plateau region, demonstrating a totally different ion transport pattern. Assuming that a single SEI was formed on the surface as described in typical theories, both $R_{ct}$ and $R_{SEI}$ should be basically the same throughout the discharging process, which is similar to the case of $Li^+$ storage in graphite anodes. This observation is clearly at odds with the experimental results. However, by introducing the dual-SEI model, these inconsistencies can be effectively resolved. For the slope region corresponding to $Na^+$ adsorption or insertion, the $Na^+$ is provided by the bulk electrolyte and transports through the S-SEI of hard carbons. Whereas for pore filling region, the solvated $Na^+$ in the nanopore undergoes complete desolvation on the I-SEI, and then forms Na clusters filled in the nanopore. The pre-desolvation that occurs on the nanopore results in a higher aggregation, which effectively reduces the $R_{ct}$. At the same time, it generates a thinner SEI with improved inorganic content, which typically has a smaller $R_{SEI}$ than the organic-dominated SEI. Therefore, both the $R_{ct}$ and $R_{SEI}$ of the pore-filling region are significantly lower than that of the insertion/adsorption region. For the charging process, the $R_{SEI}$ increases sharply when transitioning from the pore filling to the insertion/adsorption region (Supplementary Fig. 16), which may result in the $Na^+$ being trapped inside the hard carbon after fully charging. $Na^+$ is transported through different SEIs in slope and plateau regions. Note that, although $Na^+$ has different storage mechanisms in hard carbons when using ether and ester electrolytes. Their charge and discharge curves are indeed highly overlapping, indicating that using Na metal as a reference electrode in two electrode systems will not lead to differences in overpotentials. Therefore, we can use Na‖hard carbon coin cell to analyze the difference in $Na^+$ storage behavior in two different electrolytes. By controlling the discharge depth, the desolvation process on the S-SEI and I-SEI can be separated. Maintaining the hard carbon at a fully charged and discharged state, respectively, the activation energy of desolvation ($E_{a1}$) on S-SEI and I-SEI can be tested (Fig. 4h, i). The $E_{a1}$ on the S-SEI formed in ether electrolytes is 21.33 kJ mol$^{-1}$. It is much larger than the I-SEI (14.54 kJ mol$^{-1}$). For the case using ester electrolytes, the $E_{a1}$ is also much smaller for the I-SEI (69.85 kJ mol$^{-1}$ vs 51.25 kJ mol$^{-1}$). The reduced activation energy for the I-SEI can be attributed to the pre-desolvation on the nanopore, which greatly reduces the activation energy for final complete desolvation. Apart from the reduced $E_{a1}$, the activation energy for $Na^+$ transport through the SEI ($E_{a2}$) is also much lower for the I-SEI than the S-SEI whether in ether or ester electrolytes. Obviously, the formation of dual-SEI is an important reason for the reduced ICE. However, the composition and relative amount of the dual-SEI remain unclear, making it difficult to evaluate the contribution to ICE loss. In addition, whether there are other factors that affect ICE is currently uncertain. There are spatial differences in S-SEI and I-SEI distribution, and the relevant SEI components have different binding energies, which allow us to determine their relative content by XPS with depth profiling.

Since the SEIs formed in ether and ester electrolytes are significantly different in compositions and properties, we performed a comparative analysis of cycled hard carbons (after 10 cycles) in both electrolytes. As shown in Fig. 5a, f, with etching, the C 1s and Na 1s

peaks shift towards low and high binding energies, respectively, due to the formation of S-SEIs containing organic components with electron-withdrawing properties. The C 1s and Na 1s peaks of hard carbons cycled in ester electrolytes reach stabilization simultaneously after 12 min etching (Fig. 5e, f), suggesting that the S-SEI is completely removed. However, for the counterpart in ether electrolytes, the C 1s and Na 1s peaks reach a steady state after etching for 3.5 min. The turning points in Fig. 5e, f can be seen as the boundaries of the S-SEI and I-SEI. It is evident that the S-SEI formed in ether electrolytes is much thinner. It is consistent with the TEM observation. Apart from the difference in thickness, the composition of the S-SEI is also different. Due to the poor reductive stability of the ester electrolyte, a large amount of solvent is decomposed to form an organic-dominated SEI. Therefore, the initial C 1s peak position is 0.389 eV higher than the peak reaching stabilization (Fig. 5e). By contrast, it only shifts 0.122 eV in ether electrolytes, suggesting that the S-SEI contains fewer organic species (Fig. 5c, e). Then, it is further verified by Na 1s spectra. The Na 1s peak of hard carbon harvested from ester electrolytes shifts 1.117 eV towards lower binding energy (Fig. 5f), which is much larger than that cycled in ether electrolyte (0.170 eV). Correspondingly, distinct peaks associated with organic species of O=C–O and C=O bonds can be identified for hard carbons obtained from ester electrolytes (Fig. 5b). On the outermost surface of the SEI, organic components such as C–O, C=O, and O=C–O can be identified from the fitting results, which account for 26.6%, 9.6%, and 17.9%, respectively. It is significantly higher than that in ether electrolytes (Fig. 5d). The specific surface area of the prepared hard carbon is 3.85 m$^2$ g$^{-1}$ calculated based on the non-local density functional theory (NLDFT) method, which is close to the specific surface area of bulk commercial graphite[61,62]. The detected specific surface area is mainly contributed by the outer surface and is responsible for the formation of S-SEI. Such a low specific surface area effectively mitigates electrolyte decomposition and allows the hard carbon to achieve high ICE of over 98% after aging for 10 days in ether electrolytes. Even after the S-SEI is removed by $Ar^+$ etching, C–O and O=C–O still have high contents accounting for 33.7% and 14.4%, respectively, for hard carbon cycled in ester electrolytes, which can be attributed to the composition in the I-SEI. Because, at the same etching time, those organic-related bonds are absent in pristine hard carbons (Supplementary Fig. 10). By contrast, typical organic species in I-SEI formed in ether electrolytes is significantly lower than that formed in ester electrolytes (Fig. 5c and Supplementary Fig. 17). The C=O bond almost disappears after only 3.5 min of etching. Moreover, for the typical SEI constituent of F, the F 1s intensity of hard carbons cycled in ester electrolytes is about ten times higher than that in ether electrolytes (Fig. 5g and Supplementary Fig. 18). Therefore, both the S-SEI and I-SEI formed in ether electrolytes are much thinner and contain fewer organics, which enables better kinetics and reversibility of Na storage in ether electrolytes (Supplementary Fig. 19). In addition, the aggregation of electrolytes inside the nanopore is higher than outside. Therefore, the I-SEI is not only thinner but also has higher inorganic ingredients compared to the S-SEI (Supplementary Fig. 20), which can be verified by F 1s fitting results (Supplementary Fig. 21). The content of organic F species in the outermost layer of the S-SEI formed in the ester electrolyte is about 60%, while the I-SEI is fully composed of inorganic NaF species. For the case using ether electrolytes, the F-containing species in the S-SEI are mainly NaF, and only the outermost layer contains a small amount of organic F compounds (14.5%). For the I-SEI, the F-containing substance is completely NaF. The compositional evolution of SEI can be further revealed from the Na 1s fitting results (Supplementary Fig. 22). The surface of the S-SEI formed in ester electrolytes is mainly composed of organic components accompanied by a small amount of inorganic species. The organic and inorganic components in the S-SEI present a gradient distribution. The ionic Na (mainly NaF/Na$_2$O) species gradually increase from the surface to the inner layer, while the organic components gradually decrease. For the

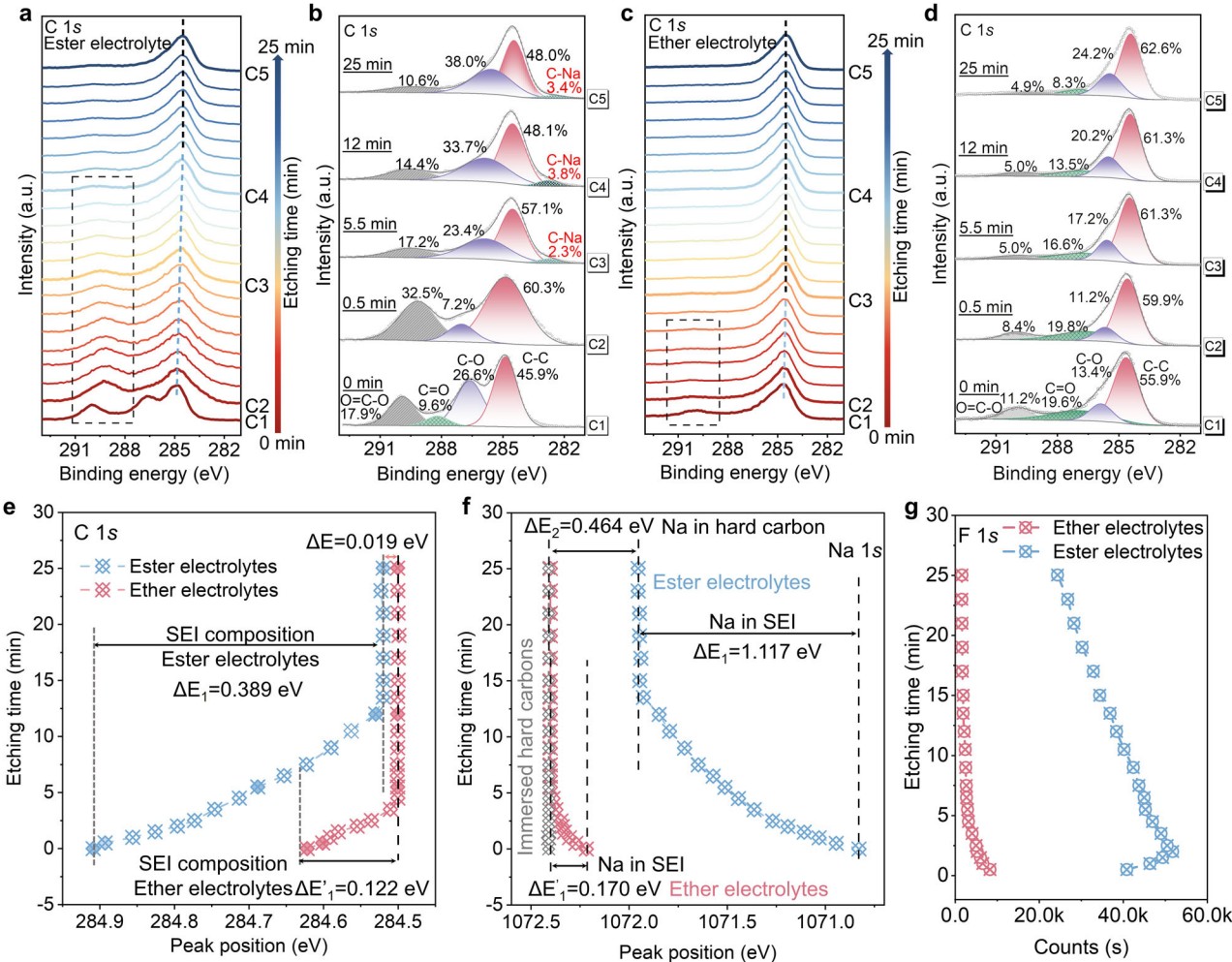

**Fig. 5 | XPS analysis of hard carbon anodes. a, b** XPS spectra of C 1s with different etching times (0–25 min) collected from the cycled hard carbon in 1 M NaPF$_6$-EC/DEC electrolytes (**a**) and corresponding fitting results (**b**) at etching time of 0, 0.5, 5.5, 12, and 25 min. **c, d** XPS spectra of C 1s with different etching times (0–25 min) collected from the cycled hard carbon in 1 M NaPF$_6$-G2 electrolytes (**c**) and corresponding fitting results (**d**) at etching time of 0, 0.5, 5.5, 12 and 25 min. **e, f** The C 1s (**e**) and Na 1s (**f**) peak position changes of cycled hard carbons with etching time. The gray line shows the Na 1s peak position changes of hard carbons immersed in ether electrolytes. **g** The F 1s signal intensity changes with etching time.

I-SEI, its composition distribution is relatively uniform and mainly dominated by NaF/Na$_2$O. But for the SEI formed in ether electrolytes, it is completely composed of NaF/Na$_2$O and uniformly distributed in both the S-SEI and the I-SEI.

In addition to compositional changes in SEIs, a new peak at about 282.5 eV can be identified from C 1s fitting results for the hard carbon cycled in ester electrolytes (Fig. 5b). This peak corresponds to the typical graphite intercalation compounds[63], suggesting that the Na$^+$ may be intercalated/inserted into the graphite microdomain of hard carbons. The insertion of Na$^+$ redistributes the charge on the hard carbon, leading to an increase in the Fermi level[64], which results in the binding energy positively shifted by 0.019 eV (Fig. 5e). Note that, this portion of Na$^+$ is irreversibly inserted since the hard carbon is in the charged state. However, for the hard carbon cycled in ether electrolytes, no C−Na bond can be detected throughout the etching process (Fig. 5d), indicating that Na$^+$ can be reversibly removed during charging. The significantly enhanced reversibility may be related to the solvent-co-intercalated storage mechanism[65]. The reversible Na$^+$ storage in ether electrolytes can be further verified by Na 1s spectra (Fig. 5f). The Na 1s core level of the hard carbon cycled in ether electrolytes returns to 1072.4 eV, which corresponds to ionic Na in immersed hard carbons as discussed before, indicating that the inserted Na$^+$ is fully extracted during charging. However, for the case in

ester electrolytes, a peak shift of 0.464 eV can be observed, demonstrating that a large amount of Na$^+$ is trapped in the graphene sheets to form intercalation compound analogs, which reduces the ionicity of Na$^+$ and lowers the binding energy. After the SEI is completely removed, hard carbons harvested in ether and ester electrolytes still retains 4.42% and 24.19% Na, respectively (Supplementary Fig. 23). It is significantly higher than the residual Na in hard carbons after immersing (less than 1%). The larger Na/C atomic ratio for hard carbon cycled in ester electrolytes also indicates that large amounts of Na$^+$ are irreversibly consumed. Due to the formation energy of Na$^+$-graphite intercalation compounds is much greater than that of equivalent Li$^+$ and K$^+$ intercalation compounds, making it difficult for Na$^+$ to be de-intercalated during charging[66–68]. Therefore, a portion of Na$^+$ is irreversibly trapped in the graphene sheets of hard carbons after forming intercalation compounds. It is known that the ester electrolyte has much higher chemical reactivity towards Na metal than ether electrolytes. Ester electrolytes would react spontaneously with Na metal to produce carbonates and release olefin gases, which results in a porous SEI and continuous decomposition of electrolytes[69,70]. The accumulation of carbonates with low ionic conductivity would lead to an increased $R_{SEI}$, resulting in poor cycling stability in Na metal batteries. However, the solid carbonates cannot shuttle, and the olefin gases cannot react with the hard carbon. Therefore, the chemical reaction

between the electrolyte and the Na anode does not significantly affect the properties and characterization of the SEI on hard carbons.

Based on the above analysis, we can rationally designate three pathways for irreversible Na loss as illustrated in Supplementary Fig. 24. The first pathway involves the formation of S-SEI, which constitutes a significant cause of Na⁺ loss. In addition, the formation of I-SEI also consumes a considerable amount of Na⁺. Finally, the irreversible loss due to Na⁺ insertion into graphene sheets accounts for a large proportion. In both ether and ester electrolytes, the ICE can be substantially improved by aging, which is achieved by increasing the aggregation of electrolytes in the nanopore and reducing electrolyte decomposition. However, the thick SEI formed in ester electrolytes and the resulting poor ion transport kinetics result in a large amount of Na⁺ being trapped in the hard carbon, which makes its ICE less than 90% even after sufficient aging. By contrast, in the ether electrolytes, the S-SEI is almost negligible, and its unique solvent co-intercalation mechanism makes it highly reversible, resulting in almost no Na⁺ loss during cycling. Under this premise, increasing the aggregation of the electrolyte in the nanopore by simple aging can greatly inhibit its decomposition, which can promote its ICE to a high average value of 98.21%.

## Electrochemical performance

Improving ICE by extending aging time does not come at the expense of capacities and cycling stability. After aging for 10 days, a high ICE of 98.95% is approached, and a high capacity of 319.7 mAh g⁻¹ can still be maintained after cycling for 250 cycles (about 10 months) at specific current of 20 mA g⁻¹ (Fig. 6a). The almost overlapped charge–discharge curves indicate that the hard carbon anode remains stable after aging (Fig. 6b). At higher specific current of 50 mA g⁻¹, it delivered a capacity of 304.9 mAh g⁻¹ over 300 cycles (Supplementary Fig. 25), with a capacity retention of 97.9%. For previous reports, the N/P capacity ratio is generally over 1.2 to compensate for the low ICE and ensure stable operation for full cells[25,26]. Before conducting the test, the hard carbon‖NNCFM batteries are pre-aged for 10 days at OCP. Benefiting from the high ICE, the full cell with a very low N/P capacity ratio of 1.02 can be constructed in this work. It delivers a high reversible capacity of 311.0 mAh g⁻¹ based on the anode mass, and an energy density of 282 W h kg⁻¹ is reached based on the total active materials of negative and positive electrodes (Supplementary Fig. 26). Then, a 4.7 Ah-class

pouch cell was assembled. Based on the total mass of the pouch cell, an energy density of 124 W h kg⁻¹ can be obtained (Supplementary Fig. 27), which further validates the applicability of our proposed strategy for large-scale applications. In addition to the high energy density enabled by the high ICE, the full cell exhibits excellent cycling stability. After running for 500 cycles at specific current of 0.5 and 1 A g⁻¹, the capacity retentions of hard carbon‖NNCFM full cell are 79.4% and 79.0%, respectively (Fig. 6c and Supplementary Fig. 28). Interestingly, the thickness of the S-SEI remains basically unchanged after 500 cycles according to the XPS analysis (Supplementary Fig. 29). But the content of organic species can see an increase for both S-SEI and I-SEI. The XPS tests indicate that the SEI formed in the ether electrolyte is relatively stable, which can effectively prevent the growth of SEI, but its composition would change slightly with cycling. Besides, it also exhibits good rate capability. The full cells show high capacities of 282.5 mAh g⁻¹ and 237.1 mAh g⁻¹ under specific currents of 0.2 A g⁻¹ and 1 A g⁻¹, respectively, which are significantly higher than the half cell (Supplementary Fig. 26). Besides, the high-voltage plateau capacity contribution of the full battery does not compromise with the increase of specific currents (Fig. 6d). This difference between full cells and half cells originates from the difference between plate electrodes and porous electrodes. Time-dependent electrochemical behavior and the performance difference between full cells and half cells derive from the porosity nature of hard carbons and related pre-desolvation, which is generally neglected in previous studies.

In summary, our findings point out that the generally overlooked porosity is important to understand the mechanism of Na storage in hard carbons, which enables some unexpected properties due to the spontaneous pre-desolvation induced by osmotic pressure and hydrostatic force on its nanopore. A dual-SEI model is proposed based on the pre-desolvation, which well elucidates the irreversible Na losses in hard carbon anodes by valence analysis with depth profiling. Importantly, the long-standing problem of low ICE can be resolved by simply prolonging the aging time to achieve sufficient pre-desolvation. The correlation between aging time and Na⁺ storage efficiency is established by introducing nanopore-based pre-desolvation. A high ICE of 98.21% can be reached without sacrificing stability and lifespan. Therefore, a full cell with a low N/P ratio of 1.02 is successfully constructed and contributes to a high energy density of 282 Wh kg⁻¹. The electrolyte and associated interfacial chemistry generated by nano

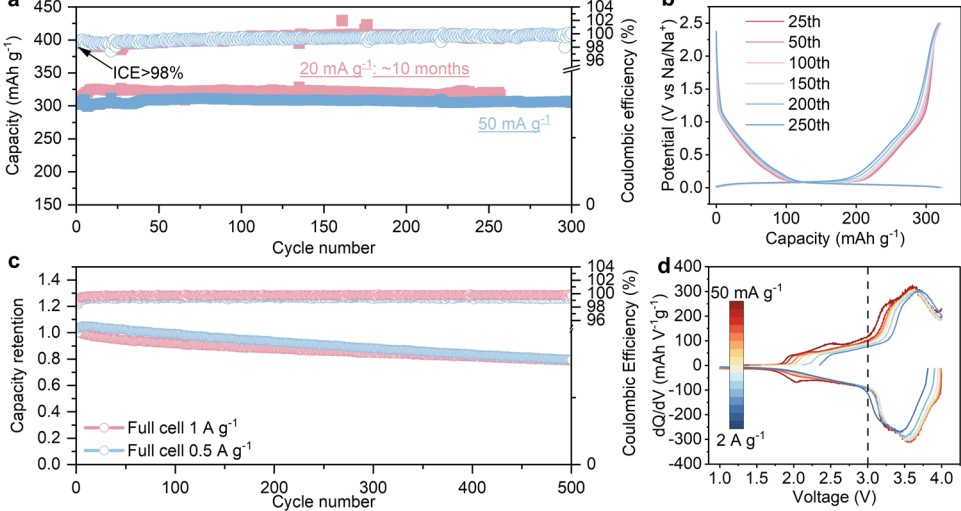

**Fig. 6 | Electrochemical performances of hard carbons. a** Cycling stability of hard carbon anodes in ether electrolytes at specific currents of 20 mA g⁻¹ and 50 mA g⁻¹ after aging for 10 days. **b** Selected charge-discharge curves at a specific current of 20 mA g⁻¹. **c** Cycling stability of hard carbon‖NNCFM full cells. The initial discharge capacities at 0.5 A g⁻¹ and 1 A g⁻¹ are 252.6 mAh g⁻¹ and 251.7 mAh g⁻¹, respectively. **d** dQ/dV curves of full cells at specific currents of 0.05−2 A g⁻¹.

confining complement the understanding of the $Na^+$ transport and storage in hard carbons, which would facilitate the design of more practical hard carbons and promote its commercial application.

## Methods

### Preparation of hard carbon anode

The hard carbon was synthesized according to previous reports[14,46]. Typically, 5 g glucose is dissolved in 45 mL of deionized water. It was then transferred to a 100 mL autoclave reactor vessel and reacted for 12 h at 230 °C. The solid product was isolated by filtration and washed three times sequentially with deionized water and absolute ethanol. The obtained solid was dried under a vacuum at 80 °C overnight. The resulting sample was then carbonized at 1400 °C for 2 h under Argon atmospheres. The hard carbon anode was prepared by mixing the hard carbon powder with acetylene black and sodium alginate (SA) In a ratio of 9:0.5:0.5, and water was used as a solvent. The fully stirred sticky slurry was coated on a Cu foil using a scraper. Then, the electrode was vacuum-dried at 100 °C overnight. For coin-cell measurements, the electrodes were punched into a plate with a diameter of 10 mm. The $O_3-NaCu_{1/9}Ni_{2/9}Fe_{1/3}Mn_{1/3}O_2$ (NNCFM) positive electrode used in full cell evaluation, was prepared by mixing the NNCFM powder with super P and PVDF in a ratio of 8:1:1 using NMP as solvents and the final sticky slurry was coated on an Al foil using a scraper. Then, the NNCFM-coated Al foil was vacuum-dried at 100 °C for 24 h. Finally, the electrodes were punched into a plate with a diameter of 10 mm for full-cell tests.

### Cell assembly and electrochemical measurements

CR2032 coin cells were used to evaluate the battery performance. The ICE was tested through Na Hard carbon half cells at different aging times. The diameter of the hard carbon electrode is 10 mm with a mass loading of ~0.9 mg cm$^{-2}$. The Na metal anode has a thickness of over 200 μm with a diameter of 12 mm (the capacity of Na metal far exceeds that of hard carbon). Typical Celgard 2500 was used as the separator with a diameter of 16 mm and the amount of electrolyte was controlled at 30 μL. The Na hard carbon half-cell was tested in the potential range of 0.005–2.5 V vs $Na^+$/Na. The same conditions were used to test the cycling stability of the hard carbon anode after resting. The Na-ion full cell was assembled using hard carbon as the negative and NNCFM as the positive electrode. The weight ratio of negative to positive was 1:2.42 (N/P capacity ratio = 1.02). The mass loading of the hard carbon anode in the full cell is about 0.65 mg cm$^{-2}$ and 1.57 mg cm$^{-2}$ for the NNCFM positive electrode. Here, the diameter of both positive and negative electrodes is the same (10 mm). Similar to the case in the half cell, Celgard 2500 was used as the separator, and 30 μL electrolytes were added in the full cell. The full cells were tested in a voltage range of 1–4 V. The aging process of a full cell is similar to that of a half cell. The hard carbon‖NNCFM full cells are aged for 10 days at OCP before conducting the test. All batteries were assembled inside the glove box with both oxygen and water contents below 1.0 ppm. The 1 M $NaPF_6$-G2 electrolyte and 1 M $NaPF_6$-EC/DEC electrolytes were used as electrolytes. Neware battery test system (Neware Technology Co.) and HJ1001SD8 (Hokuto Denko) battery test system were used to evaluate the rate and cycling stability of half and full cells. The EIS was collected in a CHI 660E electrochemical workstation, and the EIS was measured in the frequency range of 10 mHz–100 kHz with a PC signal amplitude of 10 mV. All tests were performed at room temperature (25 °C). The specific currents shown in the text are based on the mass of the hard carbon anode. The pouch cell was assembled using the layered transition metal oxides as the positive and the hard carbon as the negative. The positive and negative active materials are coated on both sides of the Al foil current collector. The loading capacity of the positive electrode is 38 mg cm$^{-2}$ and the loading capacity of the negative electrode is 15.5 mg cm$^{-2}$. The active area was 8.8 × 6.5 cm$^{-2}$ with 28-folds. A single Al tab is used in the pouch cell for both positive and negative electrodes. Note that, ester electrolytes are used in pouch cells since ether electrolytes are unstable at high voltages, especially at such a high loading. For the formation process, the cell is first charged at 0.05 C for 2 h, then charged at 0.1 C for 1 h, and finally charged at 0.2 C for 2.5 h.

### Characterization

Scanning electron microscopy was performed on a Hitachi S4800 (Hitachi Japan) instrument to analyze the morphology of cycled hard carbon anodes. X-ray diffraction measurements were performed on a Bruker D8 Advanced diffractometer fitted with Cu Kα (λ = 1.5406 Å) radiation. Elemental analysis was performed on an X-ray photoelectron spectrometer (XPS, Thermo ESCALAB 250 Xi, Al Kα radiation, $hv$ = 1486.6 eV, America). Before XPS tests, the cycled hard carbon was washed with dimethoxyethane three times to remove residual electrolytes and then dried by evaporation in a vacuum chamber. The hard carbon anode was pasted on the loading chamber with a vacuum transfer function. In the process of transfer, the sample is sealed in the loading chamber and always protected by an Ar or vacuum atmosphere. The electrode is not exposed to any reactive atmosphere during the whole process. For the XPS tests of the immersed hard carbon electrode, to avoid the influence of F and Na elements due to the introduction of binders (PVDF, PTFE, and SA), F- and Na-free binder polyacrylic acid was used in this situation. For the XPS tests of cycled hard carbon electrodes, F-free SA was used as binders. Note that, we did not calibrate all C 1s data to 284.8 eV. Because the shift of peaks would be covered up during the etching process if all data were calibrated based on standard C−C bond. After the final etching, the C 1s peak of hard carbon is closer to the standard C−C bond. Therefore, we perform peak calibration on the XPS data after the last etching and use this as a benchmark to calibrate the data collected from the previous etching process. TEM observations were performed on a JEM-1400Flash (Japan). The morphology and mechanical behavior of the outer SEI on hard carbons were recorded in an AFM using tapping mode imaging with an antimony-doped silicon tip (BRUKER RTESPA-300-30). $N_2$ adsorption−desorption at 77 K and $CO_2$ adsorption−desorption at 273 K were performed using the BELSORP MAX analyzer. The pycnometry experiments of the prepared hard carbons were conducted on BELPycno (MicrotracBEL Japan, Inc.) to detect the skeletal density. In addition, each sample was tested more than three times to ensure the accuracy of the results. The $^{23}Na$ NMR experiments with/without magic-angle-spinning were performed on a Bruker AVANCE III 400 MHz spectrometer with a 2.5 mm double-resonance HX probe. A $^{23}Na$ Hahn-echo pulse sequence (90°-τ−180°-τ) experiment was performed with a 90° pulse length of 6 μs and a delay time of 1 s. The number of scans was set to 512 to ensure quantitative analysis. The $^{23}Na$ NMR spectra were referenced to 0 ppm using NaCl as an external reference. The mass ratio of electrolytes to hard carbons in the test is 1:1.

### In situ Raman analysis on solvation structure on hard carbons

The in situ Raman spectra were performed using a JASCO microscope spectrometer (NRS-1000DT) with ×50 long-focus lens (Olympus America). A 632.8 nm excitation light from an air-cooled He−Ne laser was used in the measurement. The acquisition time was 90 s with two accumulations. In general, two sets of positions are selected for spectral acquisition and repeated twice to obtain credible Raman spectra. The spectral resolution was ~1.0 cm$^{-1}$ according to instrument settings. To avoid the solvation structure of the electrolyte due to long-term exposure to the air, the hard carbon electrode was placed in an in situ Raman cell (Hohsen Corp., Osaka, Japan). A thin quartz window with a thickness of 0.5 mm was fixed on the top of the in situ Raman cell. Raman signal was collected through the quartz window. Note that, the battery must be tightly sealed to prevent the volatilization of electrolytes from affecting the solvation structure of the electrolyte in hard carbons during the long-term resting process. In addition, the

laser is only turned on during the signal acquisition process to avoid prolonged exposure leading to increased temperatures that could affect the test results. The hard carbon electrode used here was prepared by mixing hard carbon and PVDF. Twenty microliter electrolytes were added to the hard carbon and the cell was assembled in a glovebox filled with Ar gas.

## The calculation of activation energy

The activation energy can be calculated based on the law of Arrhenius.

$$k = Ae^{-E_a/RT}$$

where $k$ is the reaction rate constant, $T$ is the temperature, $E_a$ is the activation energy, $R$ is the molar gas constant, and $A$ is the pre-exponential factor.

The law of Arrhenius in indefinite integral form:

$$lnk = \frac{-E_a}{RT} + lnA$$

$lnk$ has a linear relationship with $1/T$, and the slope is $-E_a/R$. Therefore, $E_a$ can be calculated by fitting the linear relationship between $lnk$ and $1/T$.

For electrochemical reactions, the reciprocal of the impedance value can be used in place of the rate constant for fitting. Before conducting EIS tests, the battery was cycled for 5 cycles to form a stable SEI. Its EIS was then tested over a series of gradient temperatures. It should be noted that the test temperature range should not be too wide, otherwise the fitting will deviate from the linear relationship. The impedance for desolvation ($R_1$) and Na$^+$ transport through SEI ($R_2$) can be obtained by fitting the Nyquist plot. The Arrhenius relation has a linear relation between $ln(1/R)$ and $1/T$. By fitting the linear relationship, the activation energy $E_a$ can be finally obtained.

## Theoretical calculations

The MD calculations were carried out with Materials Studio 2020. At first, all molecules were optimized geometrically in the DMol3 module. The convergence tolerances were 1.0 e-5 for energy, 0.002 Ha/Å for max. force, and 0.005 Å for max. displacement. The SCF tolerance was 1.0 e-6. The functional theory was Perdew–Burke–Ernzerhof (PBE) and generalized gradient approximation (GGA). Then, MD simulations were carried out in the amorphous cell module. The 1 M electrolyte snapshot contained 560 G2, 80 Na$^+$, and 80 PF$_6^-$. The 3.2 M electrolyte snapshot contained 560 G2, 255 Na$^+$, and 255 PF$_6^-$. The force field was COMPASIII. All the snapshots were conducted at the NPH pattern with a coupling time of 0.5 ns until the lattice length remained stable. After that, the snapshots were carried out at the NVT pattern for 2 ns to the equilibrated state. The simulated step was 1 fs. Each simulation was repeated at least two times to ensure the accuracy of the results.

All the DFT calculations were carried out using the projector augmented wave (PAW) potentials and the PBE functional[71–73] implemented in the Vienna ab initio simulation package (VASP)[74,75]. All the structures were optimized using the Gaussian smearing with SIGMA = 0.05 eV, and the energy convergence was selected $1 \times 10^{-5}$ eV atom$^{-1}$. The van der Waals (vdW) interaction was taken into account by using the vdW-DF2 functional. The transition state geometry was searched through NEB calculation. A carbon ring with a diameter of 4 Å is modeled as a hard carbon pore entrance to simulate the diffusion resistance of different solvation structures through it. In the whole process, six different diffusion positions are selected, and each position is searched for its transition state to determine a reasonable path.

## Data availability

All data is available in the main text or the supplementary information. Source data are provided with this paper.

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

## Acknowledgements

The work was funded by the Science and Technology Innovation Fund for emission peak and carbon neutrality of Jiangsu province (BK20220034).

## Author contributions

Z.L. and H.Z. conceived the idea. Z.L. synthesized the hard carbon, prepared electrolytes, and conducted related electrochemical tests. Y.G. performed all the XPS tests. H.L. and P.S. conducted an NMR test. Z.L. and H.Y. contributed new reagents/analytic tools. Z.L., H.Y., P.H., S.W., Y.Y., Q.-H.Y. and H.Z. analyzed data. Z.L., Q.-H.Y. and H.Z. prepared the draft manuscript.

## Competing interests

The authors declare no competing interests.
