## [Peer Review File · Nature Communications]

Consummating ion desolvation in hard carbon anodes for reversible sodium storageReviewer #1 (Remarks to the Author):

This study deals with the understanding of the Na storage mechanism in hard carbons for Na-ion batteries and discusses especially about the Na⁺ desolvation occurring when the electrolyte fills the nanopores.

The paper is well written and the study is well conducted, but should be submitted elsewhere as it lacks the scientific impact expected for Nature Communications.

In addition, the following comments must be considered:

- First of all, such desolvation process is already largely reported for microporous carbons used for supercapacitors applications. See for example, J. Am. Chem. Soc. 2015, 137, 39, 12627. Please modify your introduction,
- you have to change the title of the paper as you cannot give two digits on your ICE value since your error bar is much higher (standard deviation is 1.55% as reported in the text). In addition, the paper is rather devoted to the understanding of the Na⁺ desolvation process and the resulting effect on the Na storage mechanism. Please change the title accordingly,

The following remarks must be also considered before a future submission to another journal:

- Page 3, line 40: you wrote that the ICE is typically below 80 % in conventional carbonate electrolytes, which is completely untrue since many papers report ICE exceeding these values,
- Page 4, line 80: the reference is missing for the sentence "which is significantly higher than the recently reported ICE data"
- The calculation of the energy density of the full cell, based only on the weights of active materials, is useless. You must take into consideration weight of electrolyte, binder, current collectors..
- The simulation of the diffusion barrier was performed for a diameter of 4 Å. The choice of this value must be discussed in the text.
- Page 12, line 251: The sentence "Whereas for pore filling region, the Na⁺ is provided by pre-desolvated electrolytes stored in nanopores and transports through the I-SEI formed on its wall" must be reformulated as it is not clear enough,
- Page 16, line 334: it is not explained why the Na⁺ insertion into graphene sheets is irreversible for the ester-based electrolytes. There is no reason for this reaction being irreversible,
- Figure 6: you cannot compare the cycling stability of half cell and full cell as it is well known that the SEI on metallic sodium has a poor stability and therefore leads to a fast capacity decay upon cycling,
- You should perform He pycnometry experiments on your hard carbons and discuss about the kinetics of He diffusion for (ultra-)micropores filling,
- Page 19, line 395: what is the reason for using Cu as current collector, as one of the main advantages of Na-ion batteries is the possible use of Al instead,
- The galvanostatic profile of your positive NNCFM electrode must be presented somewhere,
- The carbon loading of your electrodes is very low (0.6-0.9 mg/cm²) and very far from applications. How is actually affected the ICE and your electrochemical performances for thicker electrodes?

Reviewer #2 (Remarks to the Author):

This is a very interesting and important work in understanding the interfacial chemistry generated by nanopore-based pre-desolvation, which complements the Na⁺ transport and storage mechanism in hard carbons. The dual-SEI model proposed in this work is very impressive, which well explains the irreversible Na⁺ loss during the initial cycles. The relationship between ICE and pre-desolvation is presented and verified by in situ and ex situ analysis. This study is well-designed, and the results are convincing. For these considerations, I believe that this paper is adequate to be published in Nature Communications after a proper revision by addressing the

following comments and suggestions:

1. Regarding the analysis of Raman and FTIR of electrolytes, some parts lack reference support, such as the description of symmetric P-F stretching in Figure 3a. Please provide relevant references. In addition, experimental details of in situ Raman for studying the evolution of the solvation structure are suggested to provide.
2. In Figure 1b, the aging time corresponding to charge and discharge curves is missing. It is recommended to add relevant descriptions.
3. The activation energy test methods and related details should be provided for repeatability by interested researchers. In addition, how to calculate the activation energy (E_a)? Detailed information, including the equations and fitting methods, should be supplemented.
4. XPS depth profile tests provide a lot of useful and interesting information, which was rarely reported in previous studies. What aspects of the test need to be taken into account in order to obtain reliable results? And what should be paid attention to in the processing of XPS data?
5. What does the gray line in Figure 5f represent? It is recommended to clarify in the figure or figure caption.

Reviewer #3 (Remarks to the Author):

This work have identified the time-dependent ion pre-desolvation on the nanopore of hard carbons, which significantly affects the Na⁺ storage efficiency by altering the solvation structure of electrolytes. By extending the aging time, the pre-desolvation can be consummated, generating a highly aggregated electrolyte configuration inside the nanopore, resulting in negligible reductive decomposition of electrolytes. The performance are excellent, the related mechanism will give guidance to the development Na ion batteries. Therefore, we recommend to publish this manuscript on Nature Communications after minor revision, the correspondent comments are shown below:

1. Exploiting suitable theoretical models from the perspective of atomic interactions, investigators can maximize the effects of reaction conditions. The first thorny problem is how to construct a realistic model that depicts experimentally-determined characteristics (i.e. process-structure relationship)?

Your work in Supplementary Figure 3 utilized graphene ring models to represent hard carbon electrodes, which are far to capture the complexity of local structure environments (especially extensive curved structures). Although these representations are helpful, there has been a challenge in relating them to real systems. These literature may be of value to you.

[1] Wang Y C, Zhu Y B, Wu H A. Formation and topological structure of three-dimensional disordered graphene networks[J]. Physical Chemistry Chemical Physics, 2021, 23(17): 10290-10302.

[2] Li J, Li T, Peng C, et al. Molecular structure evaluation and image-guided atomistic representation of hard carbon electrodes[J]. Journal of The Electrochemical Society, 2022, 169(7): 070517.

2. The argumentation of the authors about the structural differences is based on TEM images. I do not think that a 2D image of a 3D material allows for such profound conclusions. It is difficult to do more but the structure models presented are not based on a particular statistic method, more by the eye of the authors. Is there any possibility to go for TEM in higher magnification and then apply a statistical model over larger areas such as Pre et al have been proposing (10.1016/j.carbon.2012.09.026, 10.1149/1945-7111/ac8e36 or something related)? As now, there is only point analysis. The differences from Raman and XRD seem insignificant to me. The differences to ordinary hard-carbons ("house of cards") are not as obvious as authors are claiming.

3. More information of the S-SEI and I-SEI should be given.

4. The surface area should also be taken into account for SEI formation.

5. What is the state of SEI on hard carbon after 500 cycles in full cell?

6. In the comparison table of ICE (Table S1), the type of hard carbon should be mentioned.

7. Does this method work for commercial hard carbon?

Manuscript: NCOMMS-23-23095-T

Title: “Consummating ion desolvation in hard carbon anodes for reversible sodium storage with 99.09% initial Coulombic efficiency”

The authors greatly appreciate reviewers’ insightful comments and careful review on our manuscript (NCOMMS-23-23095-T). This paper has been revised carefully according to the comments of the reviewers. The responses are listed point-by-point in the following contents, and revisions have been highlighted by **yellow color** in the revised manuscript. Following are our responses and detailed explanation towards these comments from the reviewers.

Responses to Reviewers:

To Reviewer #1:	2-13
To Reviewer #2:	14-17
To Reviewer #3:	18-29

Response to Reviewers' comments

Reviewer #1:

General comments: This study deals with the understanding of the Na storage mechanism in hard carbons for Na-ion batteries and discusses especially about the Na⁺ desolvation occurring when the electrolyte fills the nanopores. The paper is well written and the study is well conducted, but should be submitted elsewhere as it lacks the scientific impact expected for Nature Communications. In addition, the following comments must be considered:

- First of all, such desolvation process is already largely reported for microporous carbons used for supercapacitors applications. See for example, *J. Am. Chem. Soc.* **2015**, *137*, *39*, 12627. Please modify your introduction,
- you have to change the title of the paper as you cannot give two digits on your ICE value since your error bar is much higher (standard deviation is 1.55% as reported in the text). In addition, the paper is rather devoted to the understanding of the Na⁺ desolvation process and the resulting effect on the Na storage mechanism. Please change the title accordingly,

The following remarks must be also considered before a future submission to another journal:

Response: Thank you very much for your comments and suggestions for further improving our manuscript. During the revision, a series of experiments and discussions have been performed to provide more evidence to support our results and improve the quality of this manuscript. Please also see the detailed responses to the following comments. Regarding the reviewers' comments on the novelty of our research, we would like to further illustrate the differences between our work and previous studies and highlight innovations of our study from following aspects.

(1) The nanopore-based pre-desolvation process and solvation structure evolution on hard carbons is identified for the first time

Actually, the dominated study about hard carbons believes that the desolvation process only occurs on the SEI formed on its surface (*Chem. Soc. Rev.* **2022**, *51*, 4484-4536; *Adv. Energy Mater.* **2022**, *12*, 2200715). Its nanopores can not be detected by N₂ adsorption-desorption tests, so the pore characteristics it possesses are generally ignored. The pore characteristics of hard carbon can only be revealed by using smaller sized CO₂ or He gas as molecular probes. Even if the pore characteristics of hard carbon are discussed, most studies associate it with electrolyte decomposition and SEI formation (*J. Electrochem. Soc.* **2020**, *167*, 070526; *Front. Chem.* **2022**, *10*, 986541), and do not discuss its relationship with desolvation. In this work, we explicitly identify the nanopore-based pre-desolvation process on hard carbons for the first time. The solvation structure evolution during desolvation was elucidated by using in-situ Raman, NMR, and molecular dynamics simulations.

At the same time, we have to admit that the desolvation process on microporous carbons was discussed in supercapacitors applications as mentioned by the reviewer. Actually, the desolvation widely exists in electrochemical processes. Any process involving electrochemical reactions or charge transfers at the electrode surface involves ion desolvation such as metal deposition, electrocatalysis, and charge-

discharge processes in batteries (*J. Comput. Chem.* **2007**, *28*, 1145-1152; *Nat. Commun.* **2022**, *13*, 172). However, the desolvation process of different electrochemical systems is very different and produces completely different effects. The desolvation process occurred on hard carbon is different from the case of microporous carbons in supercapacitors, which is reflected in the following aspects.

(a) Although previous studies have discussed that the confinement effect of nanopores would affect desolvation and associated molecular diffusion dynamics, they have not identified the transformation of solvation structures on microporous carbons (*J. Am. Chem. Soc.* **2015**, *137*, 12627-12632). In addition, the desolvation and related ion diffusion in nanoporous carbon are presented through calculations or simulations, and there is no direct spectroscopic experimental evidence. In our studies, we characterized and discussed the solvation structure evolution of solvated ions before and after pre-desolvation in detail by using in-situ Raman and NMR. The desolvation process on the nanopore of hard carbons is actually a pre-desolvation, which is different from previous studies. Solvated ions undergo pre-desolvation after through nanopores and involve into solvation structures with higher aggregation rather than forming naked ions. As shown in Fig. 3f, the solvation structure evolution from SSIPs to aggregated CIPs and AGGs was well identified. It is obviously different from the desolvation that occurs due to adsorption on microporous carbons in supercapacitors. Another important difference is that the desolvation on microporous carbon in supercapacitors is potential-dependent, whereas the pre-desolvation on hard carbons occurs spontaneously without applying additional potential, which is also the main reason why the ICE of hard carbon anode can be improved by simply prolonging the aging time.

(b) From the characteristics of the material, the pore features of microporous carbon used in supercapacitors can be tested by N₂ adsorption-desorption tests, and the performance of supercapacitors is often proportional to the accessible specific surface area (*J. Energy Chem.* **2013**, *22*, 226-240). But for hard carbons, the pore characteristics can only be revealed by using smaller sized CO₂ or He gas due to its smaller pore size, which would result in completely different desolvation process and ion conduction dynamics. In addition, the performance of hard carbon in Na-ion batteries, especially the ICE, is inversely proportional to the specific surface area. Therefore, the hard carbon in Na-ion batteries is different from the microporous carbon used in supercapacitors. The resulting conclusion in supercapacitors is also inapplicable for hard carbons in Na-ion batteries.

(c) From the perspective of energy storage mechanism, solvated ions in supercapacitors undergo desolvation at the micropores, and the charge is directly adsorbed on the surface of the micropores for charge storage. But for hard carbons in Na-ion batteries, the solvated ions are not directly adsorbed on the surface of nanopore after pre-desolvation but are further inserted into graphene sheets of hard carbons or undergo a pore filling process. There is a further complete desolvation to achieve the insertion or pore filling after the pre-desolvation on the nanopore. In other words, the performance of microporous carbon in supercapacitors is mainly related to specific surface area, while the performance of hard carbon in Na-ion batteries is mainly related

to the pore size. It should be noted that microporous carbon usually has extremely low ICE and basically no plateau region in Na-ion batteries (*Carbon* **2019**, *147*, 214-226; *RSC Adv.* **2014**, *4*, 64692-64697), which is different from the hard carbon in Na-ion batteries. This difference in electrochemical behavior originates from the difference in material structure and desolvation process. Although the pore-based desolvation occurs in both supercapacitors and Na-ion batteries, the effects and associated charge storage mechanisms are distinct.

(2) We establish the relationship between the pre-desolvation process and the ICE for hard carbon anodes.

In addition to the pore-based pre-desolvation process on hard carbons, we further reveal the impact of this process on the electrochemical performance, which significantly affects the Na⁺ storage efficiency by altering the solvation structure of electrolytes. Consummating the pre-desolvation by extending the aging time, it generates a highly aggregated electrolyte configuration inside the nanopore, resulting in negligible reductive decomposition of electrolytes. An ultra-high ICE of 99.09% can be reached through simply prolonging the aging time to achieve sufficient desolvation. We first establish the correlation between the desolvation process and the ICE. The ICE can be effectively improved by consummating the pre-desolvation through simply extending the aging time, which is industrially feasible. It is important for both basic research and practical applications.

(3) The irreversible Na⁺ loss in initial cycles is clearly elucidated

A major hurdle to the commercialization of hard carbon anodes is the low ICE. However, the irreversible loss of Na⁺ during initial cycle is not yet clear, so it is difficult to effectively and rationally design hard carbons with high ICE. Based on the detailed XPS analysis with depth etching, we explicitly elucidate the irreversible Na⁺ loss through our proposed dual-SEI model, which can be attributed to the formation of S-SEI and I-SEI, and the irreversible loss due to Na⁺ insertion into graphene sheets. Moreover, we discussed and analyzed the relative content of each component. Only after identifying the cause of Na⁺ loss, can we rationally design hard carbon materials or electrolytes to improve the ICE. This work provides guidance for the design of Na-ion batteries with high ICE.

Based on the above discussion and analysis, we believe that our work is distinguished from desolvation on microporous carbon in supercapacitors. This work contributes to the discussion and analysis of the unique ion desolvation on the nanopore of hard carbons and the resulting effect on Na⁺ storage, which makes the study sufficiently novel and innovative.

Now we have highlighted the innovativeness of our studies and illustrated the differences between our work and previous studies in the revised manuscript as follows: **Line 18-20, Page 4** “The solvation structure evolution during the pre-desolvation process has a significant impact on the reversibility of Na⁺ storage, which is distinguished from the desolvation effect on microporous carbon in supercapacitors^{24,25}.”

Line 22-23, Page 29; Line 1-3, Page 30

24. Pean C, *et al.* Confinement, desolvation, and electrosorption effects on the

diffusion of ions in nanoporous carbon electrodes. *J. Am. Chem. Soc.* **137**, 12627-12632 (2015).

25. Chmiola J, Largeot C, Taberna PL, Simon P, Gogotsi Y. Desolvation of ions in subnanometer pores and its effect on capacitance and double - layer theory. *Angew. Chem. Int. Ed.* **47**, 3392-3395 (2008).

Line 22-23, Page 11; Line 1-5, Page 12 “Any process relating to electrochemical reactions or charge transfers at the electrode surface involves ion desolvation^{54,55}, which is also discussed for microporous carbons in supercapacitors²⁵. Although it was established that pore size can affect the charge storage behavior of supercapacitors, the associated solvation evolution was not revealed^{24,25,56}. In addition, the unique ultra-microporous structure of hard carbons makes its desolvation process and effects such as the impact on ICE and SEI formation completely different from those of microporous carbon in supercapacitors.”

Line 1-6, Page 33

54. Huey R, Morris GM, Olson AJ, Goodsell DS. A semiempirical free energy force field with charge - based desolvation. *J. Comput. Chem.* **28**, 1145-1152 (2007).

55. Sheng L, *et al.* Suppressing electrolyte-lithium metal reactivity via Li⁺-desolvation in uniform nano-porous separator. *Nat. Commun.* **13**, 172 (2022).

56. Prehal C, *et al.* Quantification of ion confinement and desolvation in nanoporous carbon supercapacitors with modelling and in situ X-ray scattering. *Nat. Energy* **2**, 1-8 (2017).

Regarding the question in the title, we agree with the reviewer that giving two digits on ICE in the title is not accurate since the error bar is much higher. Combining our research focus on desolvation process and the resulting effect on the Na⁺ storage mechanism, we have revised the title in the manuscript as follows:

“Consummating ion desolvation in hard carbon anodes for reversible sodium storage”

We sincerely hope our explanation can give the reviewer a more comprehensive understanding of the innovativeness of our research and are also sincerely looking forward to getting support from the reviewer.

Comments 1: Page 3, line 40: you wrote that the ICE is typically below 80 % in conventional carbonate electrolytes, which is completely untrue since many papers report ICE exceeding these values,

Response: Thanks so much for the helpful comments. Here we originally wanted to express that the ICE for most of the hard carbon is still very low since more than 50% of researches on hard carbon still have ICE below 80%. We agree with the reviewer that many reported ICEs are higher than 80% by using optimized electrolytes and carefully designed hard carbons. In order to describe it more rigorously and avoid introducing ambiguity, we have modified the description in the revised manuscript as follows:

Line 5-7, Page 3 “However, a major hurdle to the commercialization of hard carbon anodes is the low initial Coulombic efficiency (ICE), which is much lower than the case of graphite anodes in Li-ion batteries^{4,5}.”

Comments 2: - Page 4, line 80: the reference is missing for the sentence “which is significantly higher than the recently reported ICE data”

Response: Many thanks for your reminding. We have added related references in the revised manuscript as follows:

Line 1-4, Page 5 “An ultra-high ICE of 99.09% can be reached through simply prolonging the aging time to achieve sufficient desolvation, which is significantly higher than the recently reported ICE data (generally below 92%) that using advanced hard carbons and optimized electrolytes²⁶⁻³⁴.”

Line 4-23, Page 30; Line 1-2, Page 31

26. Yin, X. et al. Enabling fast Na⁺ transfer kinetics in the whole - voltage - region of hard - carbon anodes for ultrahigh - rate sodium storage. *Adv. Mater.* **34**, 2109282 (2022).

27. Lu, Z. et al. Zinc Single - Atom Regulated Hard Carbons for High Rate and Low Temperature Sodium Ion Batteries. *Adv. Mater.* **35**, 2211461 (2023).

28. Bai, P. et al. Long cycle life and high rate sodium-ion chemistry for hard carbon anodes. *Energy Storage Mater.* **13**, 274-282 (2018).

29. He, Y., Bai, P., Gao, S. & Xu, Y. Marriage of an ether-based electrolyte with hard carbon anodes creates superior sodium-ion batteries with high mass loading. *ACS Appl. Mater. Interfaces* **10**, 41380-41388 (2018).

30. Xiao, B. et al. Lithium - pretreated hard carbon as high - performance sodium - ion battery anodes. *Adv. Energy Mater.* **8**, 1801441 (2018).

31. Yang, H., Xu, R. & Yu, Y. A facile strategy toward sodium-ion batteries with ultra-long cycle life and high initial Coulombic Efficiency: Free-standing porous carbon nanofiber film derived from bacterial cellulose. *Energy Storage Mater.* **22**, 105-112 (2019).

32. Dong, R. et al. Elucidating the mechanism of fast Na storage kinetics in ether electrolytes for hard carbon anodes. *Adv. Mater.* **33**, 2008810 (2021).

33. Morikawa, Y., Yamada, Y., Doi, K., Nishimura, S.-i. & Yamada, A. Reversible and high-rate hard carbon negative electrodes in a fluorine-free sodium-salt electrolyte. *Electrochemistry* **88**, 151-156 (2020).

34. Hou, B. H. et al. Self - supporting, flexible, additive - free, and scalable hard carbon paper self - interwoven by 1D microbelts: superb room/low - temperature sodium storage and working mechanism. *Adv. Mater.* **31**, 1903125 (2019).

Comment 3: - The calculation of the energy density of the full cell, based only on the weights of active materials, is useless. You must take into consideration weight of electrolyte, binder, current collectors...

Response: Thanks so much for the helpful comments. Calculating the energy density of a battery based on active materials is widely used whether in Li-ion batteries or Na-ion batteries¹⁻⁵, which can preliminarily evaluate the potential of a material in improving the energy density of batteries. At the same time, we also agree with the reviewer that calculating the energy density of full cells without considering the binder, electrolyte and current collector, etc. cannot truly reflect the energy density of the battery. For coin cells, when considering the electrolyte, binder and current collector, the energy density would be greatly reduced, which cannot reflect the actual application situation.

Therefore, a 4.7 Ah-class pouch cell was assembled. As shown in Supplementary Fig. 27, the ICE of the full cell is 96.48%. A discharge capacity of 4.72 Ah can be reached with an average discharge voltage of 3.05 V. Based on the overall mass (including active materials, electrolytes, binders, conductive carbons, current collectors, package and tabs) of the pouch cell, an energy density of 124 W h kg⁻¹ can be achieved. The energy density can be further improved by optimizing the components of pouch cells. We have supplemented the data and related discussions about pouch cells in the revised manuscript as follows:

Line 19-22, Page 19 “Then, a 4.7 Ah-class pouch cell was assembled. Based on the total mass of the pouch cell, an energy density of 124 W h kg⁻¹ can be obtained (Supplementary Fig. 27), which further validates the applicability of our proposed strategy for large-scale applications.”

We sincerely hope our explanation can satisfy the reviewer on this question and are also sincerely looking forward to getting support from the reviewer.

References

1. Suo L, *et al.* Fluorine-donating electrolytes enable highly reversible 5-V-class Li metal batteries. *Proc. Natl. Acad. Sci. U.S.A.* **115**, 1156-1161 (2018).
2. Kim U-H, *et al.* Heuristic solution for achieving long-term cycle stability for Ni-rich layered cathodes at full depth of discharge. *Nat. Energy* **5**, 860-869 (2020).
3. Qi Y, *et al.* Scalable room-temperature synthesis of multi-shelled Na₃ (VOPO₄)₂F microspheres cathodes. *Joule* **2**, 2348-2363 (2018).
4. Yin X, *et al.* Enabling fast Na⁺ transfer kinetics in the whole - voltage - region of hard - carbon anodes for ultrahigh - rate sodium storage. *Adv. Mater.* **34**, 2109282 (2022).
5. Zhang J-N, *et al.* Trace doping of multiple elements enables stable battery cycling of LiCoO₂ at 4.6 V. *Nat. Energy* **4**, 594-603 (2019).

Supplementary Figure 27 | Pouch cell performance. (a) Cycling performance of pouch cell. (b) Selected charge-discharge curves.

Comment 4: - The simulation of the diffusion barrier was performed for a diameter of 4 Å. The choice of this value must be discussed in the text.

Response: Many thanks to the reviewer’s valuable comment. To accurately confirm the pore size of hard carbons, 3A, 4A zeolite molecular sieve and zeolitic imidazolate framework-7 (ZIF-7) with well-defined pore size were tested by CO₂ adsorption

measurements. As shown in Fig. 2c, all four materials show type I adsorption isotherms, demonstrating a typical micropore feature (*Adsorpt. Sci. Technol.* **2004**, *22*, 773-782). The ZIF-7, 3A and 4A zeolite molecular sieve show almost coincided adsorption and desorption isotherms, demonstrating that there is no capillary condensation phenomenon. But for the hard carbon, it shows a H2 type hysteresis loop, which is a typical feature of ink-bottle-like pores. This kind of pore has greater filling resistance than a straight pore, taking longer time to wet and fill the nanopore. Therefore, it has a pore size of ~ 4 Å, but its CO₂ adsorption capacity is lower than 4A zeolite molecular sieve. In addition, the TEM was used to confirm the pore size. As shown in Supplementary Fig. 5, we selected two regions to test the interlayer spacing/pore size, and the average values are 0.395 and 0.405 nm, respectively, which are basically consistent with the interlayer spacing calculated from the Bragg equation (0.397 nm) (Supplementary Fig. 2). Based on the above analysis, we selected the 4 Å pore to conduct the simulation. We have added related discussion in the revised manuscript as follows:

Line 17-23, Page 6; Line 1-9, Page 7 “To accurately confirm the pore size of hard carbons, 3A, 4A zeolite molecular sieve and zeolitic imidazolate framework-7 (ZIF-7) with well-defined pore size were also tested by CO₂ adsorption measurements (Fig. 2c). All four materials show type I adsorption isotherms, demonstrating a typical micropore feature³⁵. The ZIF-7 and 3A zeolite molecular sieve have effective pore size of 2.9 Å and 3.2 Å, respectively, which are much smaller than 4 A zeolite molecular sieve. The ZIF-7 and 3A and 4A zeolite molecular sieve show almost coincided adsorption and desorption isotherms, demonstrating that there is no capillary condensation phenomenon. But for the hard carbon, it shows a H2 type hysteresis loop, which is a typical feature of ink-bottle-like pores. This kind of pore has greater filling resistance than a straight pore, taking longer time to wet and fill the nanopore. Therefore, it has a pore size of ~ 4 Å, but its CO₂ adsorption capacity is lower than 4A zeolite molecular sieve. Besides, the TEM was further used to confirm the pore size. As shown in Supplementary Fig. 5, two regions were selected to calculate the interlayer spacing/pore size, and the average values are 3.94 and 4.05 Å, respectively. It is basically consistent with the interlayer spacing calculated from the Bragg equation (0.397 nm). Therefore, the pore size of hard carbon is about 4 Å, which was used to simulate the desolvation process.”

Line 16-18, Page 8 “Since the pore size of hard carbon is the key factor affecting desolvation process^{41,42}, a graphene ring with diameter of 4 Å was applied for simplifying simulations and calculations (Fig. 2e).”

Supplementary Figure 5 | TEM analysis of pristine hard carbons. (a) TEM image of pristine hard carbon. (b) enlarged region cycled in Figure a. (c) Corresponding intensity profiles obtained from the line in Figure b. (d) enlarged region cycled in Figure a. (e) Corresponding intensity profiles obtained the line in Figure d.

Comment 5: - Page 12, line 251: The sentence “Whereas for pore filling region, the Na⁺ is provided by pre-desolvated electrolytes stored in nanopores and transports through the I-SEI formed on its wall” must be reformulated as it is not clear enough,

Response: Thanks for your suggestions. To make it clearer, we have reformulated the sentence in the revised manuscript as follows:

Line 3-5, Page 14 “Whereas for pore filling region, the solvated Na⁺ in the nanopore undergoes complete desolvation on the I-SEI, and then forms Na clusters filled in the nanopore.”

Comment 6: - Page 16, line 334: it is not explained why the Na⁺ insertion into graphene sheets is irreversible for the ester-based electrolytes. There is no reason for this reaction being irreversible,

Response: Thanks so much for your valuable comments. For Na⁺, it cannot be stably intercalated into graphite to form a stable graphite intercalation compound due to the size mismatch between Na⁺ and the graphite interlayer spacings, which is completely different from the case of Li⁺ and K⁺ (*Phys. Chem. Chem. Phys.* **2019**, *21*, 19378-19390; *RSC Adv.* **2017**, *7*, 36550-36554). Therefore, graphite can be used as the anode for Li-ion and K-ion batteries but not for Na-ion batteries. The formation energy of Na⁺-graphite intercalation compounds is much greater than that of equivalent Li⁺ and K⁺ intercalation compounds, making it difficult for Na⁺ to be de-intercalated from graphite layers during charging (*Proc. Natl. Acad. Sci. U.S.A.* **2016**, *113*, 3735-3739; *J. Phys. Chem. C* **2014**, *118*, 16-19). It is also the case for stacked graphene sheets since it has a similar structure to graphite. Therefore, the graphene sheet has poor reversibility in storing Na⁺. A portion of Na⁺ is irreversibly trapped in the graphene sheets after the initial discharge to form intercalation compounds. We have added related discussion in the revised manuscript as follows:

Line 10-14, Page 18 “Due to the size mismatch between Na⁺ and the graphite

interlayer spacings, the formation energy of Na⁺-graphite intercalation compounds is much greater than that of equivalent Li⁺ and K⁺ intercalation compounds, making it difficult for Na⁺ to be de-intercalated during charging⁶²⁻⁶⁴. Therefore, a portion of Na⁺ is irreversibly trapped in the graphene sheets of hard carbons after forming intercalation compounds.”

Line 18-23, Page 33; Line 1, Page 34

62. Liu Y, Merinov BV, Goddard III WA. Origin of low sodium capacity in graphite and generally weak substrate binding of Na and Mg among alkali and alkaline earth metals. *Proc. Natl. Acad. Sci. U.S.A.* **113**, 3735-3739 (2016).

63. Moriwake H, Kuwabara A, Fisher CA, Ikuhara Y. Why is sodium-intercalated graphite unstable? *RSC Adv.* **7**, 36550-36554 (2017).

64. Lenchuk O, Adelhelm P, Mollenhauer D. New insights into the origin of unstable sodium graphite intercalation compounds. *Phys. Chem. Chem. Phys.* **21**, 19378-19390 (2019).

Comment 7: - Figure 6: you cannot compare the cycling stability of half cell and full cell as it is well known that the SEI on metallic sodium has a poor stability and therefore leads to a fast capacity decay upon cycling,

Response: Thanks so much for your valuable comments. We have to admit that the reviewer's comment is critical and important in understanding our original motivation for comparing the performance of half cells and full cells. In ester electrolytes, the gas evolution on the Na metal anode is very severe and the formed SEI is indeed very unstable, which would cause rapid capacity decay during cycling (*ACS Appl. Mater. Interfaces* **2018**, *10*, 15270-15280; *Chem. Commun.* **2019**, *55*, 14375-14378; *Angew. Chem. Int. Ed.* **2018**, *57*, 734-737). In this case, we fully agree with the reviewer's comments that comparing the performance of full cells and half cells is unreasonable. However, the electrolyte we used in both half cells and full cells is 1 M NaPF₆-G2 ether electrolyte, which is extremely stable towards Na metal anodes. It can exhibit ultra-high average Coulombic efficiency (CE) of 99.9% in Na|Cu half cells (*ACS Cent. Sci.* **2015**, *1*, 449-455), which is comparable to the CE of hard carbon anodes. The ether electrolytes with high CE can even be used to build stable anode-free Na batteries (*Angew. Chem. Int. Ed.* **2022**, *61*, e202200410; *Nat. Energy* **2022**, *7*, 511-519). It ensures that the stability of the anode electrode is not a factor affecting the stability of half and full cells. Therefore, we compared the performance of half cells and full cells in our manuscript. Relevant discussions and statements may have been lacked in previous manuscript that led to misunderstandings by reviewers. We are very sorry for our negligence. Related discussions have been supplemented in the revised manuscript as follows:

Line 5-9, Page 20 “Note that, 1 M NaPF₆-G2 ether electrolytes were used in both half cells and full cells. The ether electrolyte enables an ultra-high CE of 99.9% for Na metal anodes⁶⁶, which is comparable to the CE of hard carbon anodes. It ensures that the stability of the anode electrode is not a factor affecting the stability of half and full cells.”

Line 4-5, Page 34

66. Seh ZW, Sun J, Sun Y, Cui Y. A highly reversible room-temperature sodium metal

anode. *ACS Cent. Sci.* **1**, 449-455 (2015).

We sincerely hope our explanation can satisfy the reviewer on this question and are also sincerely looking forward to getting support from the reviewer.

Comment 8: - You should perform He pycnometry experiments on your hard carbons and discuss about the kinetics of He diffusion for (ultra-)micropores filling,

Response: Thanks so much for your helpful comments and suggestions. He pycnometry experiments of our hard carbons were conducted on BELPycno (MicrotracBEL Japan, Inc.) to detect the skeletal density (or true density). In order to ensure the accuracy of the experiment, we conducted more than three parallel tests. The hard carbon used in our studies was prepared at 1400 °C. It shows a high skeletal density of 2.183 g cm⁻³ probed by He pycnometry experiments, which is slightly lower than the natural graphite (2.26 g cm⁻³), indicating that only a small amount of internal, inaccessible porosity exists. Note that, He gas cannot pass through sp² carbon honeycomb sheets since the He pycnometry density of graphite is 2.26 g cm⁻³ (*Chem. Mater.* **2020**, *32*, 2961-2977). Considering the principle of He pycnometry, which measures the volume of He occupying the space other than the sample in a chamber, indicating that almost all (ultra-)micropores in hard carbons are accessible for He gas. The He gas can pass through and be filled in the (ultra-)micropores at room temperature. Those (ultra-)micropores observed by TEM images (Supplementary Fig. 5) that are inaccessible for N₂ and CO₂ probe molecules can be detected by He molecules with smaller size. Therefore, solvation structures with appropriate size can enter the (ultra-)micropores of hard carbons, which further validates the pore-based pre-desolvation proposed in our studies. As the skeletal density decreases with the increase of pyrolysis temperature, the Na⁺ storage capacity can see a decrease (*Chem. Mater.* **2019**, *31*, 7288-7299; *Carbon* **2023**, *208*, 216-226), suggesting that the internal, inaccessible porosity does not contribute capacity. Therefore, the accessible open pore in the hard carbon is necessary for the storage of Na⁺. We have added related discussions about the He pycnometry experiments of hard carbons in the revised manuscript as follows:

Line 10-14, Page 7 "The hard carbon exhibits a high skeletal density of 2.183 g cm⁻³ probed by He pycnometry experiments, which is slightly lower than the natural graphite (2.26 g cm⁻³)³⁶, indicating that only a small amount of internal, inaccessible porosity exists. Therefore, those (ultra-)micropores that are inaccessible for N₂ and CO₂ probe molecules can be detected by He molecules with smaller size. It is directly related to subsequent ion desolvation."

Line 5-7, Page 31

36. Kubota K, et al. Structural analysis of sucrose-derived hard carbon and correlation with the electrochemical properties for lithium, sodium, and potassium insertion. *Chem. Mater.* **32**, 2961-2977 (2020).

Line 16-19, Page 24 "He pycnometry experiments of the prepared hard carbons was conducted on BELPycno (MicrotracBEL Japan, Inc.) to detect the skeletal density. In addition, each sample was tested more than three times to ensure the accuracy of the results."

Comment 9: - Page 19, line 395: what is the reason for using Cu as current collector, as one of the main advantages of Na-ion batteries is the possible use of Al instead,

Response: Thanks a lot for the helpful comments. As the reviewer commented, one of the advantages of Na-ion batteries is that the Al foil can be used as the current collector for the anode. However, Al is less conductive than Cu. In addition, Al collectors are easily oxidized in air. When the oxide passivation layer is formed on the surface of the Al foil, the conductivity will deteriorate¹⁻². In order to avoid large polarization caused by the internal resistance of current collectors, we choose the Cu current collector with better conductivity and less susceptible to the formation of passivation layers. Using Cu current collectors can better reflect the intrinsic electrochemical behavior of evaluated materials and is widely used in the study of anode materials for Na-ion batteries³⁻⁷. For large-scale production of Na-ion batteries, Al current collectors have advantages over Cu current collectors from the perspective of reducing costs and increasing energy densities.

References

1. Myung S-T, Hitoshi Y, Sun Y-K. Electrochemical behavior and passivation of current collectors in lithium-ion batteries. *J. Mater. Chem.* **21**, 9891-9911 (2011).
2. Jeong H, Jang J, Jo C. A review on current collector coating methods for next-generation batteries. *Chem. Eng. J.* **446**, 136860 (2022).
3. Yin X, *et al.* Enabling fast Na⁺ transfer kinetics in the whole - voltage - region of hard - carbon anodes for ultrahigh - rate sodium storage. *Adv. Mater.* **34**, 2109282 (2022).
4. Meng Q, Lu Y, Ding F, Zhang Q, Chen L, Hu Y-S. Tuning the closed pore structure of hard carbons with the highest Na storage capacity. *ACS Energy Lett.* **4**, 2608-2612 (2019).
5. Xu Z-L, *et al.* Tailoring sodium intercalation in graphite for high energy and power sodium ion batteries. *Nat. Commun.* **10**, 2598 (2019).
6. Xie F, *et al.* Screening heteroatom configurations for reversible sloping capacity promises high - power Na - ion batteries. *Angew. Chem. Int. Ed. Engl.* **61**, e202116394 (2022).
7. Li Q, *et al.* Sieving carbons promise practical anodes with extensible low-potential plateaus for sodium batteries. *Natl. Sci. Rev.* **9**, nwac084 (2022).

Comment 10: - The galvanostatic profile of your positive NNCFM electrode must be presented somewhere,

Response: Thanks so much for the suggestions. We have added the galvanostatic profile of NNCFM cathode in supplementary materials as follows:

Supplementary Figure 28 | Selected galvanostatic profiles of half cells using NNCFM cathodes.

Comment 11: - The carbon loading of your electrodes is very low (0.6-0.9 mg/cm²) and very far from applications. How is actually affected the ICE and your electrochemical performances for thicker electrodes?

Response: Many thanks to the constructive comment and valuable questions. In order to verify the influence of thick electrodes on ICE and electrochemical performance, a thick electrode with mass loading of 13.7 mg cm⁻² was prepared, which can meet the requirement for building practical pouch cells (*J. Power Sources* **2018**, *407*, 173-179; *J. Electrochem. Soc.*, **2023**, *170*, 070512). Thick hard carbon electrodes are prepared in exactly the same way as conventional electrodes. As shown in Supplementary Fig. 3, an average ICE of 88.39% can be achieved after resting for 12 h. With prolonged resting time, the ICE can see a gradual increase, which is consistent with the trend observed at lower mass loadings. As the aging time increases to 10 days, a high average ICE of 97.17% can be reached, indicating that boosting ICE by resting is also applicable for hard carbons with high mass loading. We have added related discussion in the revised manuscript as follows:

Line 1-3, Page 6 “For thick electrodes with high mass loading (13.7 mg cm⁻²), the ICE can also see an increase with aging time (Supplementary Fig. 3). After aging for 10 days, an average ICE of 97.17% can be reached.”

Supplementary Figure 3 | a, The curve of ICE versus resting time for hard carbons with high mass loading. b, The initial charge-discharge curve at different resting time.

Reviewer #2:

General comments: This is a very interesting and important work in understanding the interfacial chemistry generated by nanopore-based pre-desolvation, which complements the Na⁺ transport and storage mechanism in hard carbons. The dual-SEI model proposed in this work is very impressive, which well explains the irreversible Na⁺ loss during the initial cycles. The relationship between ICE and pre-desolvation is presented and verified by in situ and ex situ analysis. This study is well-designed, and the results are convincing. For these considerations, I believe that this paper is adequate to be published in Nature Communications after a proper revision by addressing the following comments and suggestions:

Response: Thank you very much for your positive comments and kind suggestions for further improving our manuscript. During the revision, a series of experiments and discussions have been performed to provide more evidence to support our results and improve the quality of this manuscript. Please also see the detailed responses to the following comments.

Comment 1: Regarding the analysis of Raman and FTIR of electrolytes, some parts lack reference support, such as the description of symmetric P-F stretching in Figure 3a. Please provide relevant references. In addition, experimental details of in situ Raman for studying the evolution of the solvation structure are suggested to provide.

Response: Many thanks to the reviewer's valuable comments and suggestions. We have added related discussions in the revised manuscript as follows:

Line 14-16, Page 9 "As shown in Fig. 3a, the peak located at about 740 cm⁻¹ can be attributed to the symmetric P-F stretching of PF₆⁻ (A1g)^{47,48}, which can be observed for all electrolytes apart from the pure solvent."

Line 7-11, Page 32

47. Morales D, Ruther RE, Nanda J, Greenbaum S. Ion transport and association study of glyme-based electrolytes with lithium and sodium salts. *Electrochimica Acta* **304**, 239-245 (2019).

48. Åvall G, Wallenstein J, Cheng G, Gering KL, Johansson P, Abraham DP. Highly concentrated electrolytes: electrochemical and physicochemical characteristics of LiPF₆ in propylene carbonate solutions. *J. Electrochem. Soc.* **168**, 050521 (2021).

Line 4-5, Page 10 "The function of hard carbon in converting solvation structures can also be observed from CH₂ rocking and C-O-C stretching of G2 solvents^{49,51} as shown in Supplementary Fig. 7."

Line 12-13, Page 32; Line 17-19, Page 32

49. Han, S.-D. et al. Solvate structures and computational/spectroscopic characterization of LiPF₆ electrolytes. *J. Phys. Chem. C* **119**, 8492-8500 (2015).

51. Geysens P, et al. Solvation structure of sodium bis (fluorosulfonyl) imide-glyme solvate ionic liquids and its influence on cycling of Na-MNC cathodes. *J. Phys. Chem. B* **122**, 275-289 (2018).

The experimental details about in situ Raman tests are also supplemented in the revised manuscript as follows:

Line 11-18, Page 25 “A thin quartz window with thickness of 0.5 mm was fixed on the top of the in-situ Raman cell. Raman signal was collected through the quartz window. Note that, the battery must be tightly sealed to prevent the volatilization of electrolytes from affecting the solvation structure of the electrolyte in hard carbons during the long-term resting process. In addition, the laser is only turned on during the signal acquisition process to avoid prolonged exposure leading to increased temperatures that could affect the test results. The hard carbon electrode used here was prepared by mixing hard carbon and PVDF. 20 μL electrolytes were added on the hard carbon and the cell was assembled in glovebox filled with Ar gas.”

Comment 2: In Figure 1b, the aging time corresponding to charge and discharge curves is missing. It is recommended to add relevant descriptions.

Response: Thanks so much for the reviewer's reminding. The meaning of each curve has been labeled in Fig. 1b as shown below. The charge-discharge curves shown in Fig. 1b correspond to the associated resting times.

Fig. 1 | b, The initial charge-discharge curve at different resting time.

Comment 3: The activation energy test methods and related details should be provided for repeatability by interested researchers. In addition, how to calculate the activation energy (E_a)? Detailed information, including the equations and fitting methods, should be supplemented.

Response: Many thanks to the helpful comments and valuable suggestions. The activation energy of desolvation (E_{a1}) and Na^+ transport through the SEI (E_{a2}) can be calculated based on the law of Arrhenius.

$$k = Ae^{-E_a/RT}$$

where k is reaction rate constant, T is the temperature, E_a is the activation energy, R is the molar gas constant and A is the pre-exponential factor.

After transformation, we can get the following equation:

$$\ln k = \frac{-E_a}{RT} + \ln A$$

$\ln k$ has a linear relationship with $1/T$, and the slope is $-E_a/R$.

For electrochemical reactions, the reciprocal of the impedance value can be used in place of the rate constant for fitting. The impedance for desolvation step (R_1) and Na^+ transport through SEI (R_2) can be obtained by fitting the Nyquist plot. Here, $\ln(1/R)$ is linearly related to $1/T$ and the E_a can be calculated. We have added detailed information for testing and calculating the activation energy in the revised manuscript

as follows:

Line 19-22, Page 25; Line 1-11, Page 26 “The activation energy can be calculated based on the law of Arrhenius.

$$k = Ae^{-E_a/RT}$$

where k is reaction rate constant, T is the temperature, E_a is the activation energy, R is the molar gas constant and A is the pre-exponential factor.

The law of Arrhenius in indefinite integral form:

$$\ln k = \frac{-E_a}{RT} + \ln A$$

$\ln k$ has a linear relationship with $1/T$, and the slope is $-E_a/R$. Therefore, E_a can be calculated by fitting the linear relationship between $\ln k$ and $1/T$.

For electrochemical reactions, the reciprocal of the impedance value can be used in place of the rate constant for fitting. Before conducting EIS tests, the battery was cycled for 5 cycles to form a stable SEI. Its EIS was then tested over a series of gradient temperatures. It should be noted that the test temperature range should not be too wide, otherwise the fitting will deviate from the linear relationship. The impedance for desolvation (R_1) and Na^+ transport through SEI (R_2) can be obtained by fitting the Nyquist plot. The Arrhenius relation has a linear relation between $\ln(1/R)$ and $1/T$. By fitting the linear relationship, the activation energy E_a can be finally obtained.

Comment 4: XPS depth profile tests provide a lot of useful and interesting information, which was rarely reported in previous studies. What aspects of the test need to be taken into account in order to obtain reliable results? And what should be paid attention to in the processing of XPS data?

Response: Many thanks to the constructive comment and valuable questions. For the XPS test, the most important thing is to ensure that the hard carbon electrode is not contaminated during sample preparation and testing. Therefore, the disassembly of the battery and the transfer of the hard carbon electrode must be carried out under inert gas or vacuum. Here, a loading chamber with vacuum transfer function was used to protect the hard carbon. Although hard carbon is not as reactive as Na metal, it can still be contaminated when exposed to air. Besides, hard carbon electrode for XPS testing cannot be used with F-containing binders, which is usually ignored. Since F is a common element in SEI, it may affect the analysis of the SEI composition on the hard carbon. The samples used for comparison should be tested in the same batch to avoid binding energy correction errors. For XPS data processing, we did not calibrate all C 1s data to 284.8 eV. If all the etching data were calibrated based on standard C-C bond, the peak shift would be covered up during the etching process, which is obviously unreasonable. After the final etching, the C 1s peak position of hard carbon is closer to the standard C-C bond. Therefore, we perform peak calibration on the XPS data after the last etching and use this as a benchmark to calibrate the data collected from previous etching process. We have added related information about XPS testing and data processing in the revised manuscript as follows:

Line 4-12, Page 24 “For the XPS tests of immersed hard carbon electrode, to avoid

the influence of F and Na elements due to the introduction of binders (PVDF, PTFE and SA), F- and Na-free binder polyacrylic acid was used in this situation. For the XPS tests of cycled hard carbon electrode, F-free SA was used as binders. Note that, we did not calibrate all C 1s data to 284.8 eV. Because the shift of peaks would be covered up during the etching process if all data were calibrated based on standard C-C bond. After the final etching, the C 1s peak of hard carbon is closer to the standard C-C bond. Therefore, we perform peak calibration on the XPS data after the last etching and use this as a benchmark to calibrate the data collected from previous etching process.”

Comment 5: What does the gray line in Figure 5f represent? It is recommended to clarify in the figure or figure caption.

Response: Many thanks for your careful comments and helpful suggestions. The gray line shows the Na 1s peak position changes of hard carbons immersed in ether electrolytes, which is consistent with the information shown in Supplementary Fig. 10e. We have clarified the meaning of gray line in Fig. 5f and figure caption as follows:

Fig. 5 | f, The Na 1s peak position changes of cycled hard carbons with etching time. The gray line shows the Na 1s peak position changes of hard carbons immersed in ether electrolytes.

Reviewer #3:

General comments: this work have identified the time-dependent ion pre-desolvation on the nanopore of hard carbons, which significantly affects the Na⁺ storage efficiency by altering the solvation structure of electrolytes. By extending the aging time, the pre-desolvation can be consummated, generating a highly aggregated electrolyte configuration inside the nanopore, resulting in negligible reductive decomposition of electrolytes. The performance are excellent, the related mechanism will give guidance to the development Na ion batteries. Therefore, we recommend to publish this manuscript on Nature Communications after minor revision, the correspondent comments are shown below:

Response: Thank you very much for your positive comments and kind suggestion for further improving our manuscript. During the revision, a series of experiments and discussions have been performed to provide more evidence to support our results and improve the quality of this manuscript. Please also see the detailed responses to the following comments.

Comment 1: Exploiting suitable theoretical models from the perspective of atomic interactions, investigators can maximize the effects of reaction conditions. The first thorny problem is how to construct a realistic model that depicts experimentally-determined characteristics (i.e. process-structure relationship)? Your work in Supplementary Figure 3 utilized graphene ring models to represent hard carbon electrodes, which are far to capture the complexity of local structure environments (especially extensive curved structures). Although these representations are helpful, there has been a challenge in relating them to real systems. These literature may be of value to you.

[1] Wang Y C, Zhu Y B, Wu H A. Formation and topological structure of three-dimensional disordered graphene networks[J]. *Physical Chemistry Chemical Physics*, 2021, 23(17): 10290-10302.

[2] Li J, Li T, Peng C, et al. Molecular structure evaluation and image-guided atomistic representation of hard carbon electrodes[J]. *Journal of The Electrochemical Society*, 2022, 169(7): 070517.

Response: Thank you very much for your careful comments and providing valuable literatures, which help us to conduct deeper analysis. We fully agree with the reviewer's comment that building realistic models is crucial to rationally depict experimental phenomena. However, the amorphous structural features of hard carbon make it difficult to accurately capture and model the microstructure as discussed in the references provided by the reviewers (*Phys. Chem. Chem. Phys.* **2021**, 23,10290-10302; *J. Electrochem. Soc.* **2022**, 169, 070517). Therefore, there are many controversies about the structural characteristics of hard carbons. The pores and structural features of hard carbons can be reflected to a certain extent by constructing a three-dimensional amorphous graphene network. However, it requires the construction of very large unit cells to simulate. Taking the research of Jiaqi Li *et al.* as an example (*J. Electrochem. Soc.* **2022**, 169, 070517), a large-scale atomistic representation (C₄₈₀₂₅H₁₈₅₇O₈₁₁N₁₉₈S₁₂₇) in a 100 × 100 × 100 Å cubic cell was constructed to simulate the structure of hard carbon based on HRTEM image analysis

in combination with LDIMS, FTIR, XPS, XRD, SAXS tests. The constructed structure well illustrates the structural characteristics of hard carbons. It is more convincing to use such a model to simulate the desolvation process of the electrolyte on the nanopore of hard carbons. However, calculating the transition states of different solvation structures through nanopores of hard carbons using a model containing tens of thousands of atoms would be very computationally intensive. Besides, we believe that the pore size is a more important parameter than the pore structure in the desolvation process according to previous reports (*Joule* **2020**, *4*, 1776-1789; *Proc. Natl. Acad. Sci. U.S.A.* **2022**, *119*, e2210203119). Therefore, a simplified graphene ring with diameter of 4 Å (Fig. 2e) was applied to calculate the transition states of different solvation structures passing through the nanopore. Building models by grasping the key influencing factors to ensure its rationality and mitigate computational effort is widely used (*J. Power Sources* 2012, **198**, 329-337; *Nature* 2021, **592**, 225-231; *Joule* 2018, **2**, 2117-2132). At the same time, we have to admit that what the reviewers have pointed out here is important and forward-looking for the study of hard carbons. Using a finer structure for the simulation may yield more accurate simulation results as the reviewer commented. We will continue to study the pore and structural characteristics of hard carbon, and then establish optimized models for simulations and calculations. We have added related discussions in the revised manuscript as follows:

Line 14-20, Page 8 "To accurately reflect the structure of hard carbon, it is often necessary to establish a very fine and large-scale model^{39,40}. However, it is usually time-consuming and computationally intensive. Since the pore size of hard carbon is the key factor affecting desolvation process^{41,42}, a graphene ring with diameter of 4 Å was applied for simplifying simulations and calculations (Fig. 2e). The method of constructing models by grasping the key influencing factors to ensure its rationality while mitigating computational effort is widely used⁴³⁻⁴⁵."

Line 13-23, Page 31; Line 1-3, Page 32

39. Wang Y, Zhu Y, Wu H. Formation and topological structure of three-dimensional disordered graphene networks. *Phys. Chem. Chem. Phys.* **23**, 10290-10302 (2021).

40. Li J, Li T, Peng C, Li J, Zhang H. Molecular structure evaluation and image-guided atomistic representation of hard carbon electrodes. *J. Electrochem. Soc.* **169**, 070517 (2022).

41. Chang Z, Qiao Y, Deng H, Yang H, He P, Zhou H. A liquid electrolyte with desolvated lithium ions for lithium-metal battery. *Joule* **4**, 1776-1789 (2020).

42. Lu, Z. et al. Step-by-step desolvation enables high-rate and ultra-stable sodium storage in hard carbon anodes. *Proc. Natl. Acad. Sci. U.S.A.* **119**, e2210203119 (2022).

43. Dao T-S, Vyasarayani CP, McPhee J. Simplification and order reduction of lithium-ion battery model based on porous-electrode theory. *J. Power Sources* **198**, 329-337 (2012).

44. Baran MJ, et al. Diversity-oriented synthesis of polymer membranes with ion solvation cages. *Nature* **592**, 225-231 (2021).

45. Bai S, Sun Y, Yi J, He Y, Qiao Y, Zhou H. High-power Li-metal anode enabled by metal-organic framework modified electrolyte. *Joule* **2**, 2117-2132 (2018).

Comment 2: The argumentation of the authors about the structural differences is

based on TEM images. I do not think that a 2D image of a 3D material allows for such profound conclusions. It is difficult to do more but the structure models presented are not based on a particular statistic method, more by the eye of the authors. Is there any possibility to go for TEM in higher magnification and then apply a statistical model over larger areas such as Pre et al have been proposing (10.1016/j.carbon.2012.09.026, 10.1149/1945-7111/ac8e36 or something related)? As now, there is only point analysis. The differences from Raman and XRD seem insignificant to me. The differences to ordinary hard-carbons ("house of cards") are not as obvious as authors are claiming.

Response: Thanks so much for the constructive comment and providing valuable literatures, which help us to conduct deeper analysis. In order to obtain more accurate and fine structural information of hard carbons, we performed HRTEM tests with higher magnification as suggested by the reviewer. As shown in Supplementary Figs. 12a and 12g, the HRTEM exhibits higher clarity and resolution in displaying the structure of hard carbons. The intelligent fringe recognition method was further used to identify structural characteristics of hard carbons according to previous studies (*Carbon* **2013**, 52, 239-258; *J. Electrochem. Soc.* **2022**, 169, 090522; *J. Electrochem. Soc.* **2022**, 169, 070517). The resulting binary image can be obtained after filtering, which shows clearer fringe compared to the original HRTEM image (Supplementary Figs. 12a and 12b). Finally, the skeletonized fringe image was obtained after branch pruning, which can be used for data statistics and analysis. For the case cycled in ether electrolytes, the interlayer spacing is mainly concentrated around 0.4 nm (Supplementary Fig. 12e), which is closed to the data obtained from original HRTEM images measured by Gatan DigitalMicrograph software (Supplementary Fig. 12d). This further confirms the reliability of the statistical methods used in our studies. For the fringe length, it is mainly distributed around 0.77 nm, and there is a strong peak near 1.5 nm, indicating that it has a higher proportion for long fringe length. The same method was used to process HRTEM images of hard carbon cycled in ester electrolytes (Supplementary Figs. 12g-12i). The final skeletonized fringe image also shows clear interlayer spacing and fringe length. Compared with the hard carbon cycled in ether electrolytes, the hard carbon harvested from ester electrolytes shows much larger interlayer spacing (about 0.42 nm), which is closed to the average interlayer spacing measured by Gatan DigitalMicrograph software (Supplementary Fig. 12j). Besides, the hard carbon cycled in ester electrolytes exhibits shorter fringe length compared to the case using ether electrolytes. The increased interlayer spacing and reduced fringe length in ester electrolytes can be attributed to the irreversible intercalation of Na⁺ in hard carbon, which expands the spacing between graphene sheets and interrupts original continuous layered structure. The above results indicate that the irreversible loss of Na⁺ in hard carbon may be an important reason for low ICE, which is consistent with our subsequent XPS analysis. Interestingly, the interlayer spacing of the hard carbon is basically unchanged after repeated cycling in ether electrolytes (Supplementary Fig. 5), which may be related to its unique solvent co-intercalation mechanism and improves the reversibility of Na⁺ storage. We have added related discussions in the revised manuscript as follows:

Line 14-21, Page 12 "In addition to the difference in SEIs, significant differences can

also be seen for the structure of the hard carbon harvested from different electrolytes. As shown in Supplementary Fig. 12, the hard carbon cycled in ester electrolytes shows much larger interlayer spacing and shorter fringe length compared to the case using ether electrolytes, which originates from the irreversible intercalation of Na⁺ expanding the graphene sheets and interrupting the original continuous layered structure of hard carbons. The irreversible Na⁺ intercalation and SEI formation are responsible for the low ICE in ester electrolytes, which was confirmed in the subsequent XPS analysis.”

Supplementary Information, Line 1-11, Page 14 “In order to obtain more accurate and fine structural information from HRTEM images, the intelligent fringe recognition method was applied to identify structural characteristics of hard carbons according to previous studies¹⁰⁻¹². The resulting binary image after the filtering shows clearer fringe compared to the original HRTEM image. Finally, the skeletonized fringe image can be obtained after branch pruning, which was used for data statistics and analysis. For the case cycled in ether electrolytes, the interlayer spacing is mainly concentrated around 0.4 nm (Supplementary Figure 12e), which is much smaller than the case using ester electrolytes. For the fringe length, it is mainly distributed around 0.77 nm, and there is a strong peak near 1.5 nm, which is longer than the hard carbon harvested from ester electrolytes. The increased interlayer spacing and reduced fringe length can be attributed to the irreversible intercalation of Na⁺ in hard carbon, which expands the graphene sheets and interrupts the original continuous layered structure.”

Supplementary Information, Line 24-32, Page 35

10. Pré P, Huchet G, Jeulin D, Rouzaud J-N, Sennour M, Thorel A. A new approach to characterize the nanostructure of activated carbons from mathematical morphology applied to high resolution transmission electron microscopy images. *Carbon* **52**, 239-258 (2013).

11. Li J, Ouyang H, Wang J, Li J, Zhang H. Nanostructure Quantification of Hard Carbon Electrodes through Advanced HRTEM Image Analysis. *J. Electrochem. Soc.* **169**, 090522 (2022).

12. Li J, Li T, Peng C, Li J, Zhang H. Molecular structure evaluation and image-guided atomistic representation of hard carbon electrodes. *J. Electrochem. Soc.* **169**, 070517 (2022).

Supplementary Figure 12 | TEM analysis of hard carbons cycled in different electrolytes. (a) The HRTEM image of hard carbon harvested from ether electrolytes and corresponding preprocessed image. (b) Corresponding binary image. (c) The skeletonized fringe image. (d) The intensity profiles obtained from the line in Figure a. (e) Interlayer spacing distribution statistics for Figure c. (f) Fringe length distribution statistics for Figure c. (g) The HRTEM image of hard carbon harvested from ester electrolytes and corresponding preprocessed image. (h) Corresponding binary image. (i) The skeletonized fringe image. (j) The intensity profiles obtained from the line in Figure a. (k) Interlayer spacing distribution statistics for Figure c. (l) Fringe length distribution statistics for Figure i.

Comment 3: More information of the S-SEI and I-SEI should be given.

Response: Many thanks to the helpful comments and suggestions. More information about the S-SEI and I-SEI can be obtained by fitting F 1s and Na 1s spectra of cycled hard carbons. As shown in F 1s fitting results (Supplementary Fig. 21), the outermost layer of the S-SEI formed in the ester electrolyte consists of both organic and inorganic F species. Organic F is the dominant component, accounting for about 59.9%. But for the case using ether electrolytes, NaF is the main component of the S-SEI with a content of 85.5%. After etching for 5.5 min, the organic components disappear and only NaF remains for the S-SEI formed in both ester and ether electrolytes. The F-containing species in the S-SEI are mainly NaF, and only the outermost layer contains a small amount of organic F compounds. For the I-SEI, the F-containing substance is

completely NaF whether for the case using ester or ether electrolytes. Note that, the atomic ratio of F for SEIs formed in ester electrolytes is 10 times higher than that formed in ether electrolytes (Supplementary Fig. 18). Therefore, although the main component of SEIs formed in both electrolytes is NaF, the content of NaF in SEIs derived from ester electrolytes is much higher than that formed in ether electrolytes. More information about the distribution and composition of components in SEIs can be revealed from the Na 1s spectra. As shown in Supplementary Fig. 22, Na species associated with organic species are the dominant component for the S-SEI formed in ester electrolytes, which accounts for 76.3%. After etching for 0.5 min, the content of organic species increases slightly to 77.5%, indicating that the surface of the S-SEI is mainly composed of organic components accompanied by a small amount of inorganic species. During the subsequent etching process, the content of ionic Na gradually increases and becomes the dominant component. The organic components can be divided into two parts including organic species in the SEI and Na⁺ intercalation compounds in hard carbons. The ionic Na can be attributed to NaF and Na₂O according to previous report (*ACS Cent. Sci.* **2015**, *1*, 449–455). Therefore, organic and inorganic species are distributed in gradients in the S-SEI, and the inorganic NaF/Na₂O species gradually increase from the surface to the inner layer, while the organic components gradually decrease. For the I-SEI, its composition distribution is relatively uniform and mainly dominated by NaF/Na₂O. But for the SEI formed in ether electrolytes, it is completely composed of NaF/Na₂O and no organic species were detected, indicating that the distribution of NaF/Na₂O in both S-SEIs and I-SEIs is very uniform. The inorganic NaF and Na₂O components are generally considered to be favorable components to form robust SEIs (*J. Power Sources* **2017**, *341*, 107-113; *J. Am. Chem. Soc.* **2021**, *143*, 2829-2837; *Angew. Chem. Int. Ed.* **2021**, *133*, 12050-12055), which allows the hard carbon to exhibit excellent stability in ether electrolytes. Based on above analysis, the composition and elemental distribution in S-SEIs and I-SEIs are better revealed, which is favorable to understand the pre-desolvation and resulting effects on the nanopore of hard carbons. We have added related discussions about the S-SEI and I-SEI in the revised manuscript as follows:

Line 21-23, Page 16; Line 1-12, Page 17 “Therefore, the I-SEI is not only thinner but also has higher inorganic ingredients compared to the S-SEI (Supplementary Fig. 20), which can be verified by F 1s fitting results (Supplementary Fig. 21). The content of organic F species in the outermost layer of the S-SEI formed in the ester electrolyte is about 60%, while the I-SEI is fully composed of inorganic NaF species. For the case using ether electrolytes, the F-containing species in the S-SEI are mainly NaF, and only the outermost layer contains a small amount of organic F compounds (14.5%). For the I-SEI, the F-containing substance is completely NaF. The compositional evolution of SEI can be further revealed from the Na 1s fitting results (Supplementary Fig. 22). The surface of the S-SEI formed in ester electrolytes is mainly composed of organic components accompanied by a small amount of inorganic species. The organic and inorganic components in the S-SEI present a gradient distribution. The ionic Na (mainly NaF/Na₂O) species gradually increase from the surface to the inner layer, while the organic components gradually decrease. For the I-SEI, its composition

distribution is relatively uniform and mainly dominated by NaF/Na₂O. But for the SEI formed in ether electrolytes, it is completely composed of NaF/Na₂O and uniformly distributed in both the S-SEI and the I-SEI.”

Supplementary Figure 21 | F 1s spectra of hard carbons. a-c, XPS spectra of F 1s with different etching time (0-25 min) collected from the cycled hard carbon in 1 M NaPF₆-EC/DEC electrolytes (a) and corresponding fitting results at etching time of 0.5 min (b) and 0 min (c). d-e, XPS spectra of F 1s with different etching time (0-25 min) collected from the cycled hard carbon in 1 M NaPF₆-G2 electrolytes (d) and corresponding fitting results at etching time of 5.5 min (e) and 0 min (f).

Supplementary Figure 18 | Comparison of element content in SEI formed in different electrolytes. a, XPS atomic concentration of various elements as a function of sputtering time for the SEI formed in 1 M NaPF₆-EC/DEC electrolytes and 1 M NaPF₆-G2 electrolytes. b, Corresponding enlarged region from 0-25%.

Supplementary Figure 22 | Na 1s spectra of hard carbons. a,b, XPS spectra of Na 1s with different etching time (0-25 min) collected from the cycled hard carbon in 1 M NaPF₆-EC/DEC electrolytes (a) and corresponding fitting results (b) at etching time of 0, 0.5, 5.5, 12 and 25 min. c,d, XPS spectra of Na 1s with different etching time (0-25 min) collected from the cycled hard carbon in 1 M NaPF₆-G2 electrolytes (c) and corresponding fitting results (d) at etching time of 0, 0.5, 5.5, 12 and 25 min.

Comment 4: The surface area should also be taken into account for SEI formation.

Response: Many thanks to the helpful comments. As shown in Fig. 2c, the hard carbon has a very low CO₂ adsorption capacity. The specific surface area calculated based on the non-local density functional theory (NLDFT) method is 3.85 m² g⁻¹, which is close to the specific surface area of bulk commercial graphite (*J. Power Sources* **1999**, 81, 312-316; *Carbon* **2018**, 134, 507-518). Therefore, we believe that the specific surface area of hard carbon is mainly contributed by the outer surface, which is responsible for the formation of S-SEI. Such a low specific surface area effectively mitigates electrolyte decomposition and allows the hard carbon to achieve high ICE of over 98% after aging for 10 days in ether electrolytes. We have added related discussions in the revised manuscript as follows:

Line 3-9, Page 16 “The specific surface area of the prepared hard carbon is 3.85 m² g⁻¹ calculated based on the non-local density functional theory (NLDFT) method, which is close to the specific surface area of bulk commercial graphite^{57,58}. The detected specific surface area is mainly contributed by the outer surface and responsible for the formation of S-SEI. Such a low specific surface area effectively mitigates electrolyte decomposition and allows the hard carbon to achieve high ICE of over 98% after aging for 10 days in ether electrolytes.”

Line 7-10, Page 33

57. Simon B, Flandrois S, Guerin K, Fevrier-Bouvier A, Teulat I, Biensan P. On the choice of graphite for lithium ion batteries. *J. Power Sources* **81**, 312-316 (1999).

58. Zhao L, Bennett J, Obrovac M. Hexagonal platelet graphite and its application in Li-ion batteries. *Carbon* **134**, 507-518 (2018).

Comment 5: What is the state of SEI on hard carbon after 500 cycles in full cell?

Response: Many thanks for your helpful comments and valuable questions. In order to study the state of SEI on hard carbon after 500 cycles in full cells, we conducted

XPS tests with depth etching. To compare with the SEI formed on hard carbon after 10 cycles, all test parameters are controlled to be the same as previous tests. As shown in Supplementary Fig. 30, with etching, the C 1s and Na 1s peaks shift towards high and low binding energies, respectively, due to the formation of S-SEIs containing organic components with electron-withdrawing properties. After etching for 3.5 min, the C 1s and Na 1s peak reach stabilization simultaneously. At the same time, those organic species are basically disappeared after etching for 3.5 min, suggesting that the S-SEI is completely removed. Note that, the S-SEI disappears after the same etching time for the case after 10 cycles, indicating that the thickness of the S-SEI formed on the hard carbon after 500 cycles in the full cell is basically the same as that after 10 cycles in half cells. However, the composition of S-SEI has changed significantly. After 10 cycles, the content of O=C-O species on the surface of S-SEI is 11.2% and it becomes 8.4% after etching for 0.5 min. But for the case after 500 cycles, the proportion of O=C-O on the surface of S-SEI is 29.3%, which increases to 36.7% after etching for 0.5 min. Although the thickness of S-SEI has not changed, the content of organic species has increased significantly. When the S-SEI is removed by Ar⁺ etching, the signal of the I-SEI can be collected. The content of organic species in I-SEI formed after 500 cycles can also see an increase compared to that after 10 cycles. The proportion of O=C-O species increases from 5% to over 7%, indicating that organic species increases with cycling for both S-SEI and I-SEI. Inorganic components are usually represented by NaF, and its distribution and content can be inferred from the F 1s spectra. As shown in Supplementary Fig. 30d, the F 1s peak position is basically unchanged, only the intensity can see a slight decrease with etching, indicating that the distribution of F-related organic species in the S-SEI and I-SEI is substantially uniform. We have added related discussions in the revised manuscript as follows:

Line 9-13, Page 20 “Interestingly, the thickness of the S-SEI remains basically unchanged after 500 cycles according to the XPS analysis (Supplementary Fig. 30). But the content of organic species can see an increase for both S-SEI and I-SEI. The XPS tests indicate that the SEI formed in the ether electrolyte is relatively stable, which can effectively prevent the growth of SEI, but its composition would change slightly with cycling.”

Supplementary Information, Line 5-16, Page 32; Line 1-3, Page 33 “With etching, the C 1s and Na 1s peaks shift towards high and low binding energies, respectively due to the formation of S-SEIs containing organic components with electron-withdrawing properties. After etching for 3.5 min, the C 1s and Na 1s peak reach stabilization simultaneously. At the same time, organic-related peaks are basically disappeared, suggesting that the S-SEI is completely removed. The thickness of S-SEI is basically unchanged since the S-SEI also disappears after 3.5 min etching for the hard carbon after 500 cycles. However, the composition of S-SEI has changed significantly. The content of O=C-O species on the surface of S-SEI (after 10 cycles) is 11.2% and it becomes 8.4% after etching for 0.5 min. But for the S-SEI formed on the hard carbon after 500 cycles, the proportion of O=C-O species on the surface is 29.3%, which increases to 36.7% after etching for 0.5 min. The content of organic

components in I-SEI can also see an increase compared to that after 10 cycles. The proportion of O=C-O species in the I-SEI increases from about 5% to more than 7%. Inorganic components are usually represented by NaF, and its distribution and content can be inferred from the F 1s spectra. The F 1s peak position is basically unchanged, only the intensity can see a slight decrease with etching, suggesting that the distribution of F-related organic species in the S-SEI and I-SEI is relatively uniform.”

Supplementary Figure 30 | XPS spectra of hard carbons after 500 cycles in full cells. a,b, XPS spectra of C 1s with different etching time (0-25 min) collected from the cycled hard carbon in 1 M NaPF₆-G2 electrolytes (a) and corresponding fitting results (b) at etching time of 0, 0.5, 5.5, 12 and 25 min. c,d, The Na 1s (c) and F 1s (d) spectra with different etching time (0-21 min).

Comment 6: In the comparison table of ICE (Table S1), the type of hard carbon should be mentioned.

Response: Many thanks for your helpful comments. The type of hard carbon has been supplemented in Supplementary table 1 as follows:

Supplementary table 1. Comparison of ICE obtained under corresponding test conditions.

ICE	The type of hard carbons	Electrolyte	Binder	Separator	Current density	Refs.
99.09	HC Microspheres (Carbonizing glucose-based precursor at 1400 °C for 2h)	1 M NaPF ₆ -G2	SA	PP	20 mA g ⁻¹	This work
77.63%	HC microspheres (Carbonizing 3-aminophenol-based precursor at 1300 °C for 4h with ZnO-assisted bulk etching)	1 M NaPF ₆ -G2	PVDF	Glass fiber	50 mA g ⁻¹	1

84%	Zn doping HC (Carbonizing 2, 4-diaminophenol -based precursor and zinc acetate dihydrate at 1300 °C for 2h)	1 M NaPF ₆ -G2	PVDF	Glass fiber	50 mA g ⁻¹	2
83.8%	HC microspheres (Carbonizing sucrose-based precursor at 1000 °C for 2h)	0.8 M NaPF ₆ -G2	SA	PP	20 mA g ⁻¹	3
85.9%	HC granular (Carbonizing chitosan at 1100 °C for 2h)	1 M NaPF ₆ -DME	SA	PP	50 mA g ⁻¹	4
92.1%	Commercial HC (Provided by Kureha)	1 M NaClO ₄ -G4	PVDF	PP	50 mA g ⁻¹	5
93%	Carbon nanofiber (Carbonizing fermentation at 1300 °C for 6h)	1 M NaOTf-G2	-	Glass fiber	200 mA g ⁻¹	6
84.93%	HC nanospheres (Carbonizing xylose-based precursor at 1200 °C for 3h)	1 M NaClO ₄ -G2	SA	Glass fiber	1000 mA g ⁻¹	7
95.0%	Commercial HC (provided by Kureha)	0.5 M NaBPh ₄ -DME	CMC	Glass fiber	20 mA g ⁻¹	8
91.2%	HC paper (Carbonizing tissue at 1300 °C for 3h)	1 M NaOTf-G2	CMC	-	20 mA g ⁻¹	9

Comment 7: Does this method work for commercial hard carbon?

Response: Many thanks for your helpful comments. The commercial hard carbon produced from petroleum pitch (Carbotron P(J) obtained from Kureha Battery Materials Japan Co. Ltd.) was used to investigate whether our proposed method is applicable. The mass loading of commercial hard carbons was controlled at 1.3 mg cm⁻². As shown in Supplementary Fig. 4, the ICE gradually improves with the increase of aging time, which is consistent with the trend observed in hard carbons prepared in our laboratory. It presents an average ICE of 90.84% after aging for 12 h. As the aging time increases to 10 days, an average ICE of 97.67% can be achieved. It is slightly lower than our prepared hard carbons, which may originate from differences in specific surface area or pore size between the two hard carbons. We have added related data and discussions in the revised manuscript as follows:

Line 3-6, Page 6 "We further investigated the case of commercial hard carbons. As shown in Supplementary Fig. 4, the ICE also gradually increases with aging time. An average ICE of 97.67% can be obtained after aging for 10 days, which further verifies the universality of the established regular."

Supplementary Figure 4 | a, The curve of ICE versus resting time for commercial hard carbons. b, The initial charge-discharge curve at different resting time.

Reviewer #1 (Remarks to the Author):

This revised manuscript could be accepted for publication in Nature Communications. I would however appreciate that the following comments are considered.

1) Your reply: "As shown in Supplementary Fig. 5, we selected two regions to test the interlayer spacing/pore size, and the average values are 0.395 and 0.405 nm, respectively, which are basically consistent with the interlayer spacing calculated from the Bragg equation (0.397 nm) (Supplementary Fig. 2). Based on the above analysis, we selected the 4 Å pore to conduct the simulation."

This sentence is very confusing. The pore size is completely different than the interlayer spacing.... By TEM and XRD, you extract the interlayer spacing value in the BSUs (Basic Structural Unit) as described in the "house of cards" model. This does not take into account any curvature of the graphene layers. Between, the BSUs, there is the presence of very small pores (i.e. ultra-micropores, with pore diameter often below 1 nm). These ultra-micropores can be closed, even to gases such as He. Definitely, these nanopores cannot be probed by He and, thus, the density determined by He pycnometry can be much lower than 2 g/cm³. Nevertheless, this closed ultra-microporosity can store sodium and can be responsible for a large part of the electrochemical capacity (especially on the plateau at low voltage).

2) Your reply: "Therefore, a simplified graphene ring with diameter of 4 Å (Fig. 2e) was applied to calculate the transition states of different solvation structures passing through the nanopore."

This representation of the pore entrance is a very simplified model. I do not think it is very useful to simulate the micro-texture/structure of hard carbons, which are ultra-microporous.

3) You wrote "due to the size mismatch between Na⁺ and the graphite interlayer spacings,"

You must remove this sentence. This is not the case. I agree that the formation energy of sodium-graphite intercalation compounds is higher than those with Li or heavy alkali metals (K, Rb, Cs), but this is definitely not related to size mismatch between Na⁺ and the graphite interlayer spacing.

4) Your reply: "Considering the principle of He pycnometry, which measures the volume of He occupying the space other than the sample in a chamber, indicating that almost all (ultra-) micropores in hard carbons are accessible for He gas."

For many hard carbons, some ultra-micropores are not accessible to He but are responsible for a significant part of the electrochemical capacity.

5) Your reply: "As the skeletal density decreases with the increase of pyrolysis temperature, the Na⁺ storage capacity can see a decrease (Chem. Mater. 2019, 31, 7288-7299; Carbon 2023, 208, 216-226), suggesting that the internal, inaccessible porosity does not contribute capacity."

I disagree. Many papers have shown that the development of the closed ultra-micropore volume (not probed by CO₂ or He) leads to an increase of the reversible electrochemical capacity (especially on the plateau at low voltage on the galvanostatic profile).

Reviewer #2 (Remarks to the Author):

Previous comments have been well addressed, so I recommend the acceptance of this manuscript.

Reviewer #3 (Remarks to the Author):

This study presents a significant contribution to the comprehension of interfacial chemistry resulting from nanopore-based pre-desolvation. It serves as a valuable complement to the existing knowledge on the Na⁺ transport and storage mechanism in hard carbons. All of the concerns raised by the individual have been appropriately resolved. The present study exhibits a commendable level of design and the outcomes obtained are highly persuasive. Based on these factors, it is my belief that the content of this manuscript meets the necessary criteria for publication in Nature Communications.

Manuscript: NCOMMS-23-23095A

Title: “Consummating ion desolvation in hard carbon anodes for reversible sodium storage”

The authors greatly appreciate your insightful comments and careful review on our manuscript (NCOMMS-23-23095A). This paper has been revised carefully according to the comments of the reviewers. The responses are listed point-by-point in the following contents, and revisions have been highlighted by **yellow color** in the revised manuscript. Following are our responses and detailed explanation towards these comments from the reviewers.

Responses to Reviewers:

To Reviewer #1:	2-6
To Reviewer #2:	7
To Reviewer #3:	7

Response to Reviewers' comments

Reviewer #1:

General comments: This revised manuscript could be accepted for publication in Nature Communications. I would however appreciate that the following comments are considered.

Response: We would like to thank the insightful feedback from the Reviewer #1 and appreciate their recommendation for publication. Following the valuable comments, we have discussed it in detail and revised our manuscript carefully as per the comments.

Comments 1: Your reply: "As shown in Supplementary Fig. 5, we selected two regions to test the interlayer spacing/pore size, and the average values are 0.395 and 0.405 nm, respectively, which are basically consistent with the interlayer spacing calculated from the Bragg equation (0.397 nm) (Supplementary Fig. 2). Based on the above analysis, we selected the 4 Å pore to conduct the simulation." This sentence is very confusing. The pore size is completely different than the interlayer spacing....

By TEM and XRD, you extract the interlayer spacing value in the BSUs (Basic Structural Unit) as described in the "house of cards" model. This does not take into account any curvature of the graphene layers. Between, the BSUs, there is the presence of very small pores (i.e. ultra-micropores, with pore diameter often below 1 nm). These ultra-micropores can be closed, even to gases such as He. Definitively, these nanopores cannot be probed by He and, thus, the density determined by He pycnometry can be much lower than 2 g/cm³. Nevertheless, this closed ultra-microporosity can store sodium and can be responsible for a large part of the electrochemical capacity (especially on the plateau at low voltage).

Response: Thanks so much for your valuable comments. We agree with the reviewer that pore size and interlayer spacing are two completely different concepts. The pore size and interlayer spacing are also generally different for hard carbons. However, some regions of stacked graphene layers can also be considered as orifices of nanopores due to the short-range ordered structural features of hard carbons. But overall, it is not reasonable to characterize pore size in terms of layer spacing. We are very sorry for the imprecise description and thank the reviewers for their careful evaluation. We have revised the discussion in the new manuscript as follows:

Line 4-5, Page 7 "Therefore, it has a pore size of ~4 Å, but its CO₂ adsorption capacity is lower than 4A zeolite molecular sieve, and the 4 Å pore was used to simulate the desolvation process."

Line 8-11, Page 7 "Note that, some regions of stacked graphene layers can also be regarded as orifices of nanopores due to the short-range ordered structural features of hard carbons, which might also be detected by CO₂ adsorption measurements."

In addition, due to the limitations of the TEM and XRD testing techniques, it does not take into account any curvature of the graphene layers as commented by the reviewer. Accurately characterizing the structure of hard carbon is still very challenging due to the coexistence of complex topological structures and amorphous structures. However, ultra-micropores, which are undetectable even with He, not only do exist but are critical for the Na⁺ storage at low voltage plateau (*Nat. Commun.* **2023**, *14*, 6024;

Adv. Funct. Mater. **2023**, 2308392; *Adv. Funct. Mater.* **2022**, 32, 2203725; *SusMat.* **2022**, 2, 357-367). Increasing the topological cavity of the carbon material facilitates the formation of more closed ultra-micropores and the plateau capacity can also see an increase at the same time (*Adv. Mater.* **2023**, 35, 2302613). However, the plateau capacity would drop suddenly due to the formation of thick graphite domains that blocks the transport of Na⁺ as the pyrolysis temperature reaches to 2000 °C (*Adv. Mater.* **2023**, 35, 2302613). We agree with this statement based on the fact that the plateau capacity in ether electrolytes is more than twice that in ester electrolytes for the hard carbon with substantial closed ultra-micropores (*Adv. Mater.* **2023**, 35, 2302613). It is well known that the hard carbon has better kinetics for Na⁺ transport in ether electrolytes due to the unique solvent co-intercalation mechanism (*Adv. Mater.* **2021**, 33, e2008810; *Nat. Commun.* **2019**, 10, 2598), which enables the ultra-micropores to store Na⁺ more effectively. We have added related discussion in the revised manuscript as follows:

Line 13-16, Page 7 “Recent studies have shown that these inaccessible closed ultra-micropores can also effectively contribute to reversible capacity, especially the plateau capacity^{37,38}, and there are differences in the storage efficiency of Na⁺ for different types of closed ultra-micropores^{39,40}.”

Line 8-16, Page 31

37. Tang Z, et al. Revealing the closed pore formation of waste wood-derived hard carbon for advanced sodium-ion battery. *Nat Commun* **14**, 6024 (2023).

38. Huang Y, et al. Rationally Designing Closed Pore Structure by Carbon Dots to Evoke Sodium Storage Sites of Hard Carbon in Low - Potential Region. *Adv. Funct. Mater.* 2308392 (2023).

39. He XX, et al. Achieving All - Plateau and High - Capacity Sodium Insertion in Topological Graphitized Carbon. *Adv. Mater.* **35**, 2302613 (2023).

40. Morikawa Y, Nishimura Si, Hashimoto Ri, Ohnuma M, Yamada A. Mechanism of sodium storage in hard carbon: an X - ray scattering analysis. *Adv. Energy Mater.* **10**, 1903176 (2020).

Comments 2: Your reply: “Therefore, a simplified graphene ring with diameter of 4 Å (Fig. 2e) was applied to calculate the transition states of different solvation structures passing through the nanopore.”

This representation of the pore entrance is a very simplified model. I do not think it is very useful to simulate the micro-texture/structure of hard carbons, which are ultra-microporous.

Response: Thanks so much for the constructive comment. We agree with the reviewer’s comment that the pore entrance model used here is a very simplified model. Actually, building realistic models is crucial to rationally depict experimental phenomena. However, the amorphous structural features of hard carbon make it difficult to accurately capture the microstructure and build models (*Phys. Chem. Chem. Phys.* **2021**, 23,10290-10302; *J. Electrochem. Soc.* **2022**, 169, 070517). There are still many controversies about the structural characteristics of hard carbons (*Mater. Today.* **2019**, 23, 87-104). Given the insufficient understanding of hard carbon structures, it is difficult to construct very convincing models. In addition, even if a reasonable model

could be constructed to simulate the micro-texture/structure of hard carbon, it would be extremely time-consuming as it contains tens of thousands of atoms. For example, Jiaqi Li *et al.* built a large-scale atomistic representation ($C_{48025}H_{1857}O_{811}N_{198}S_{127}$) in a $100 \times 100 \times 100 \text{ \AA}$ cubic cell to simulate the structure of hard carbons (*J. Electrochem. Soc.* **2022**, *169*, 070517). When such a model is used to calculate the transition states of different solvation structures through nanopores of hard carbons, the computational effort increases even further, which would be extremely computationally intensive. In our work, we believe that the pore size is a more important parameter than the pore structure in the desolvation process according to previous reports (*Joule* **2020**, *4*, 1776-1789; *Proc. Natl. Acad. Sci. U.S.A.* **2022**, *119*, e2210203119). Therefore, we used a simplified graphene ring to simulate the pore entrance and calculate the transition states of different solvation structures passing through the nanopore. Constructing models by grasping the key influencing factors to ensure its rationality and mitigate computational effort is widely used (*J. Power Sources* 2012, **198**, 329-337; *Nature* 2021, **592**, 225-231; *Joule* 2018, **2**, 2117-2132). Of course, if the structural characteristics of hard carbon can be accurately captured and used to build a model for simulations, it would be more reasonable and accurate, which may also be the original intention of the reviewer's comment. We will continue to pay attention to the research progress of hard carbon and carry out related research to obtain a more comprehensive understanding of the structure of hard carbons so that we can establish a more reasonable model for theoretical analysis.

We sincerely hope our explanation can satisfy the reviewer on this question and are also sincerely looking forward to getting support from the reviewer.

Comment 3: You wrote “due to the size mismatch between Na^+ and the graphite interlayer spacings,”

You must remove this sentence. This is not the case. I agree that the formation energy of sodium-graphite intercalation compounds is higher than those with Li or heavy alkali metals (K, Rb, Cs), but this is definitively not related to size mismatch between Na^+ and the graphite interlayer spacing.

Response: Thanks so much for the helpful comments and suggestions. Both Li^+ , which is smaller in size than Na^+ , and K^+ , which is larger in size, can form stable graphite intercalation compounds (*RSC Adv.* **2017**, *7*, 36550-36554; *Phys. Chem. Chem. Phys.* **2019**, *21*, 19378-19390). Therefore, we agree with the reviewer that it is not related to the size mismatch between Na^+ and the graphite interlayer spacings. We have removed this sentence in the revised manuscript as follows:

Line 13-15, Page 18 “Due to the formation energy of Na^+ -graphite intercalation compounds is much greater than that of equivalent Li^+ and K^+ intercalation compounds, making it difficult for Na^+ to be de-intercalated during charging⁶⁶⁻⁶⁸.”

Comment 4: Your reply: “Considering the principle of He pycnometry, which measures the volume of He occupying the space other than the sample in a chamber, indicating that almost all (ultra-) micropores in hard carbons are accessible for He gas.” For many hard carbons, some ultra-micropores are not accessible to He but are responsible for a significant part of the electrochemical capacity.

Response: The hard carbon used in this work has a high skeletal density of 2.183 g

cm⁻³ probed by He pycnometry experiments, which is slightly lower than the natural graphite (2.26 g cm⁻³), indicating that only a small amount of internal, inaccessible porosity exists. After a thorough literature research, we believe that the reviewers' comments are correct. Some ultra-micropores inaccessible to He can contribute to a high capacity especially for the plateau voltage region, which has been verified by recent studies (*Nat. Commun.* **2023**, *14*, 6024; *Adv. Funct. Mater.* **2023**, 2308392; *Adv. Funct. Mater.* **2022**, *32*, 2203725; *SusMat.* **2022**, *2*, 357). Related discussions combined with the analysis in other comments about the Na⁺ storage in closed ultra-micropores have been supplemented in the revised manuscript as follows:

Line 13-16, Page 7 “Recent studies have shown that these inaccessible closed ultra-micropores can also effectively contribute to reversible capacity, especially the plateau capacity^{37,38}, and there are differences in the storage efficiency of Na⁺ for different types of closed ultra-micropores^{39,40}.”

Comment 5: Your reply: “As the skeletal density decreases with the increase of pyrolysis temperature, the Na⁺ storage capacity can see a decrease (*Chem. Mater.* 2019, *31*, 7288-7299; *Carbon* 2023, *208*, 216-226), suggesting that the internal, inaccessible porosity does not contribute capacity.”

I disagree. Many papers have shown that the development of the closed ultra-micropore volume (not probed by CO₂ or He) leads to an increase of the reversible electrochemical capacity (especially on the plateau at low voltage on the galvanostatic profile).

Response: Thanks a lot for the insightful comments. After systematic literature study and analysis, we agree with the reviewers' comment that some internal, inaccessible porosity can indeed contribute to capacity. Increasing the closed ultra-micropore would effectively increase the reversible capacity, especially for the plateau capacity (*Nat. Commun.* **2023**, *14*, 6024; *Adv. Funct. Mater.* **2023**, 2308392; *Adv. Funct. Mater.* **2022**, *32*, 2203725; *SusMat.* **2022**, *2*, 357). For example, the increase in pyrolysis temperature would lead to the formation of closed ultra-micropore and increase in pore volume, which is favorable to improve the reversible Na⁺ storage capacity (*Energy Fuels* **2016**, *30*, 7811; *Adv. Energy Mater.* **2020**, *10*, 1903176; *Angew. Chem. Int. Ed.* **2021**, *60*, 5114). At the same time, we note that both the overall capacity and the plateau capacity decrease rapidly when the pyrolysis temperature exceeds a certain critical temperature (*Energy Environ. Sci.* **2020**, *13*, 3469; *Adv. Energy Mater.* **2019**, *9*, 1901351). These phenomena indicate that not all closed pores can contribute equivalently to the capacity. If the pyrolysis temperature is too high, the graphite wall forming the closed pores will be too thick and hinder the conduction of Na⁺ (*Adv. Mater.* **2023**, *35*, 2302613), which would result in Na⁺ being stored less efficiently or even failing to contribute capacity. Therefore, we speculate that only closed pores with accessible Na⁺ conduction channels could effectively improve the plateau capacity. We sincerely thank the reviewers for the helpful and valuable comments enabling us to have a deeper thinking and understanding about the Na⁺ storage mechanism in hard carbons. Combining the discussion and analysis in previous comments, we have added related descriptions in the revised manuscript as follows:

Line 13-16, Page 7 “Recent studies have shown that these inaccessible closed ultra-

micropores can also effectively contribute to reversible capacity, especially the plateau capacity^{37,38}, and there are differences in the storage efficiency of Na⁺ for different types of closed ultra-micropores^{39,40}.

Line 8-16, Page 31

37. Tang Z, et al. Revealing the closed pore formation of waste wood-derived hard carbon for advanced sodium-ion battery. *Nat Commun* **14**, 6024 (2023).

38. Huang Y, et al. Rationally Designing Closed Pore Structure by Carbon Dots to Evoke Sodium Storage Sites of Hard Carbon in Low - Potential Region. *Adv. Funct. Mater.* 2308392 (2023).

39. He XX, et al. Achieving All - Plateau and High - Capacity Sodium Insertion in Topological Graphitized Carbon. *Adv. Mater.* **35**, 2302613 (2023).

40. Morikawa Y, Nishimura Si, Hashimoto Ri, Ohnuma M, Yamada A. Mechanism of sodium storage in hard carbon: an X - ray scattering analysis. *Adv. Energy Mater.* **10**, 1903176 (2020).

We sincerely hope our explanation can satisfy the reviewer on this question and are also sincerely looking forward to getting support from the reviewer.

Reviewer #2:

General comments: Previous comments have been well addressed, so I recommend the acceptance of this manuscript.

Response: Thank you very much for your recognition and recommendation of our work.

Reviewer #3:

General comments: This study presents a significant contribution to the comprehension of interfacial chemistry resulting from nanopore-based pre-desolvation. It serves as a valuable complement to the existing knowledge on the Na⁺ transport and storage mechanism in hard carbons. All of the concerns raised by the individual have been appropriately resolved. The present study exhibits a commendable level of design and the outcomes obtained are highly persuasive. Based on these factors, it is my belief that the content of this manuscript meets the necessary criteria for publication in Nature Communications.

Response: We really appreciate the reviewer's support and recommendation of our work for publication in Nature Communications.